# Hepatic Sdf2l1 controls feeding-induced ER stress and regulates metabolism

Takayoshi Sasako [1,2,3,4,5], Mitsuru Ohsugi[1], Naoto Kubota[1,2,6], Shinsuke Itoh[1,7], Yukiko Okazaki[1,3], Ai Terai[1,3], Tetsuya Kubota[1,8,9], Satoshi Yamashita[10], Kunio Nakatsukasa[11,12], Takumi Kamura[11], Kaito Iwayama[13], Kumpei Tokuyama[13], Hiroshi Kiyonari[14,15], Yasuhide Furuta[14,15], Junji Shibahara[16], Masashi Fukayama[16], Kenichiro Enooku[17], Kazuya Okushin[17], Takeya Tsutsumi[18], Ryosuke Tateishi [17], Kazuyuki Tobe[19], Hiroshi Asahara[10], Kazuhiko Koike[17], Takashi Kadowaki [1,2,20,21] & Kohjiro Ueki[1,2,3]

Dynamic metabolic changes occur in the liver during the transition between fasting and feeding. Here we show that transient ER stress responses in the liver following feeding terminated by Sdf2l1 are essential for normal glucose and lipid homeostasis. Sdf2l1 regulates ERAD through interaction with a trafficking protein, TMED10. Suppression of Sdf2l1 expression in the liver results in insulin resistance and increases triglyceride content with sustained ER stress. In obese and diabetic mice, Sdf2l1 is downregulated due to decreased levels of nuclear XBP-1s, whereas restoration of Sdf2l1 expression ameliorates glucose intolerance and fatty liver with decreased ER stress. In diabetic patients, insufficient induction of Sdf2l1 correlates with progression of insulin resistance and steatohepatitis. Therefore, failure to build an ER stress response in the liver may be a causal factor in obesity-related diabetes and nonalcoholic steatohepatitis, for which Sdf2l1 could serve as a therapeutic target and sensitive biomarker.

[1] Department of Diabetes and Metabolic Diseases, Graduate School of Medicine, The University of Tokyo, Tokyo 113-8655, Japan. [2] Translational Systems Biology and Medicine Initiative (TSBMI), The University of Tokyo, Tokyo 113-8655, Japan. [3] Department of Molecular Diabetic Medicine, Diabetes Research Center, National Center for Global Health and Medicine, Tokyo 162-8655, Japan. [4] Division for Health Service Promotion, The University of Tokyo, Tokyo 113-0033, Japan. [5] Department of Molecular Sciences on Diabetes, Graduate School of Medicine, The University of Tokyo, Tokyo 113-8655, Japan. [6] Department of Clinical Nutrition Therapy, The University of Tokyo Hospital, The University of Tokyo, Tokyo 113-865, Japan. [7] Kowa Company Limited, Nagoya 460-0003, Japan. [8] Clinical Nutrition Program, National Institute of Health and Nutrition, Tokyo 162-8636, Japan. [9] Division of Cardiovascular Medicine, Toho University Ohashi Medical Center, Tokyo 143-8541, Japan. [10] Department of Systems BioMedicine, Tokyo Medical and Dental University, Tokyo 113-8510, Japan. [11] Division of Biological Sciences, Graduate School of Science, Nagoya University, Nagoya 464-8601, Japan. [12] Graduate School of Natural Sciences, Nagoya City University, Nagoya 464-8601, Japan. [13] Graduate School of Comprehensive Human Science, University of Tsukuba, Tsukuba 305-8577, Japan. [14] Animal Resource Development Unit, RIKEN Center for Life Science Technologies, Kobe 650-0047, Japan. [15] Genetic Engineering Team, RIKEN Center for Life Science Technologies, Kobe 650-0047, Japan. [16] Department of Pathology, Graduate School of Medicine, The University of Tokyo, Tokyo 113-8655, Japan. [17] Department of Gastroenterology, Graduate School of Medicine, The University of Tokyo, Tokyo 113-8655, Japan. [18] Department of Infectious Disease, Graduate School of Medicine, The University of Tokyo, Tokyo 113-8655, Japan. [19] The First Department of Internal Medicine, Graduate School of Medicine and Pharmaceutical Sciences of Research, The University of Toyama, Toyama 930-8555, Japan. [20] Department of Prevention of Diabetes and Lifestyle-Related Diseases, Graduate School of Medicine, The University of Tokyo, Tokyo 113-8655, Japan. [21] Department of Metabolism and Nutrition, Mizonokuchi Hospital, Faculty of Medicine, Teikyo University, Tokyo 213-8507, Japan. Correspondence and requests for materials should be addressed to T.K. (email: kadowaki-3im@h.u-tokyo.ac.jp) or to K.U. (email: ueki-tky@umin.net)

Glucose and lipid metabolism in the liver undergo dynamic changes during the transition between fasting and feeding[1]. During fasting, the liver releases glucose by glycogenolysis and gluconeogenesis, and ketone bodies by fatty acid oxidation, while during feeding, it stores excessive nutrition derived from food by synthesizing glycogen and fatty acids. Insulin is a major regulator in this context by promoting anabolism and suppressing catabolism[2–5].

Conversely, dysregulation of these processes may lead to metabolic disorders. For instance, we have previously shown that in obesity, hepatic IRS-2 expression during fasting, which should be up-regulated, is eventually down-regulated due to hyperinsulinemia, resulting in impaired insulin signaling in the liver[6]. Hepatic insulin resistance, in turn, accelerates hyperinsulinemia itself, which impairs insulin signaling in other tissues as well[7]. Hyperinsulinemia also contributes to up-regulation of hepatic SREBP1c even during fasting, when it should be down-regulated, causing excessive fatty acid synthesis[8,9]. However, our understanding of the dynamic metabolic regulation in the liver prompted by fasting and feeding is still limited and it remains largely unknown how dysregulation of this process causes metabolic diseases, such as type 2 diabetes.

Endoplasmic reticulum (ER) stress is becoming an emerging player in the regulation of metabolism in the liver. The ER is an organelle involved in synthesis of secretory and membrane proteins. In the ER, unfolded proteins, immediately after translation and entrance into the organelle, are matured through modification, such as folding, initiation of glycosylation, and formation of disulfide bonds. Under ER stress, in which unfolded proteins accumulate in the ER due to increased protein synthesis or chaperone dysfunction, various responses are induced, including both cytoprotective responses and cytotoxic ones[10]. In the field of metabolism, impaired or excessive responses to chronic ER stress are thought to result in hepatic insulin resistance and fatty liver disease[11–18]. There has been a controversy, however, about whether ER stress and ER stress responses are enhanced or suppressed in obesity and diabetes[19–21]. It is still unclear what stimulation induces ER stress in the liver, and which molecule mainly resolves the stress. Moreover, in humans, although some ER stress markers are elevated in insulin resistance and non-alcoholic steatohepatitis (NASH)[22,23], little is known about the contribution of ER stress responses to these disorders.

In this study, we identify an ER-resident molecule, stromal cell-derived factor 2 like 1 (Sdf2l1) as a physiological regulator of ER stress responses induced by feeding in the liver, and demonstrate that suppression of the molecule causes sustained ER stress, leading to insulin resistance and hepatic steatosis. These data reveal a crucial link between ER stress and both insulin resistance and fatty liver disease.

## Results

**Feeding induces ER stress responses in the liver**. To explore the precise mechanism and physiological implications of the dynamic metabolic changes between fasting and feeding conditions in the liver, we searched the microarray data using murine liver samples comparing the fasting and feeding conditions in the public domain, and found a data set (GEO accession: [GSE59885]), indicating 193 transcripts up-regulated (Supplementary Table 1) after refeeding in the control mice. Those up-regulated included ER stress-related genes, such as *Hspa5* (encoding BiP), *Syvn1* (encoding Hrd1), *Ero1lb*, *Dnajb11* (encoding ERdj3), *Ddit3* (encoding CHOP), *Hyou1* (encoding ORP150), *Pdia3* (encoding PDI), and *Xbp1*.

We were particularly interested in *Sdf2l1* among the genes highly up-regulated by refeeding, which showed an about 6-fold increase in expression. Sdf2l1 is thought to be an ortholog of ER proteins Pmt1p and Pmt2p, both of which are *O*-mannosyltransferases in yeast, and has been reported to have an ER-retention-like motif in the C terminus[24], which is essential for transportation from the Golgi apparatus back to the ER[25], and to function as a component of the ER chaperone complex[26–29]. Little is known, however, of its role in glucose and lipid metabolism.

We then further examined the time course in detail. Although expression of ER stress marker genes, including classical ones, namely spliced *Xbp1* (*sXbp1*, encoding XBP-1s), *Hspa5*, and *Ddit3*, as well as *Sdf2l1*, showed little change during fasting, it was elevated prominently and transiently after refeeding (Fig. 1a and Supplementary Fig. 1a). Thus we focused on the refed state thereafter, and at the protein level, phosphorylation, expression, and nuclear localization of ER stress markers, as well as Sdf2l1, were also elevated during refeeding, as analyzed by total lysates (Fig. 1b and Supplementary Fig. 1b, c), immunoprecipitates (Supplementary Fig. 1d), and nuclear extracts (Fig. 1c). Moreover, Sdf2l1, as well as BiP, was isolated mainly in microsomal fractions, which also showed elevation during refeeding (Fig. 1d). These data suggest that transient ER stress is induced in the liver by the physiological stimulation of feeding, even in lean nondiabetic mice.

We also found that *Sdf2l1*, but not the other orthologs of Pmt1/2p, was induced by feeding in the liver, but not or hardly in other tissues, despite wide expression in insulin-targeted organs (Supplementary Fig. 1a, e, f).

ER stress is usually caused in response to the increased burden of protein synthesis, and we found that some of the regulators were activated by refeeding (Supplementary Fig. 1c). Nutrients and insulin that abundantly reach the liver during feeding are known to promote protein synthesis[3,5,30]. Thus, we assessed the effect of nutrients by depleting each of them from the animals' feed (except carbohydrates, because the deprivation markedly decreased food intake itself even after 24-h fasting). This revealed that deprivation of protein partially attenuated the feeding-induced ER stress responses, whereas lipid deprivation failed, except slightly attenuated induction of *Sdf2l1* (Supplementary Fig. 2a). We also assessed the effect of insulin, using mice treated with streptozotocin (STZ) as an insulin-deficient animal model, and found that the ER stress responses were also partially suppressed by the treatment (Supplementary Fig. 2c). Finally, STZ-treated mice fed with protein-deprived feed exhibited almost complete suppression of ER stress responses during feeding (Supplementary Fig. 2e), paralleled with suppressed activation of protein synthesis markers (Supplementary Fig. 2b, d, f).

**Induction of Sdf2l1 as an ER stress response**. Although expression of Sdf2l1 in the liver was increased in parallel with the ER stress responses, it remained unclear whether ER stress directly induced the expression. Indeed, Sdf2l1 was induced by tunicamycin and thapsigargin in Fao cells (Supplementary Fig. 3a), suggesting that it might be a component of ER stress responses. Furthermore, we generated an adenovirus encoding luciferase under the *Sdf2l1* promoter, Ad-Sdf2l1-Luc, and confirmed the promoter activity was strongly enhanced both in vitro by tunicamycin and in vivo by refeeding (Supplementary Fig. 3b,c).

Next, we performed promoter assays using Fao cells, and found that the *Sdf2l1* promoter activity was enhanced by ER stress (Fig. 1e and Supplementary Fig. 3d). Deletion experiments revealed that the region within ~−100 bp is essential for the ER stress mediated activation of the promoter (Fig. 1e, f), where we found a motif similar to ER stress response elements (ERSEs)

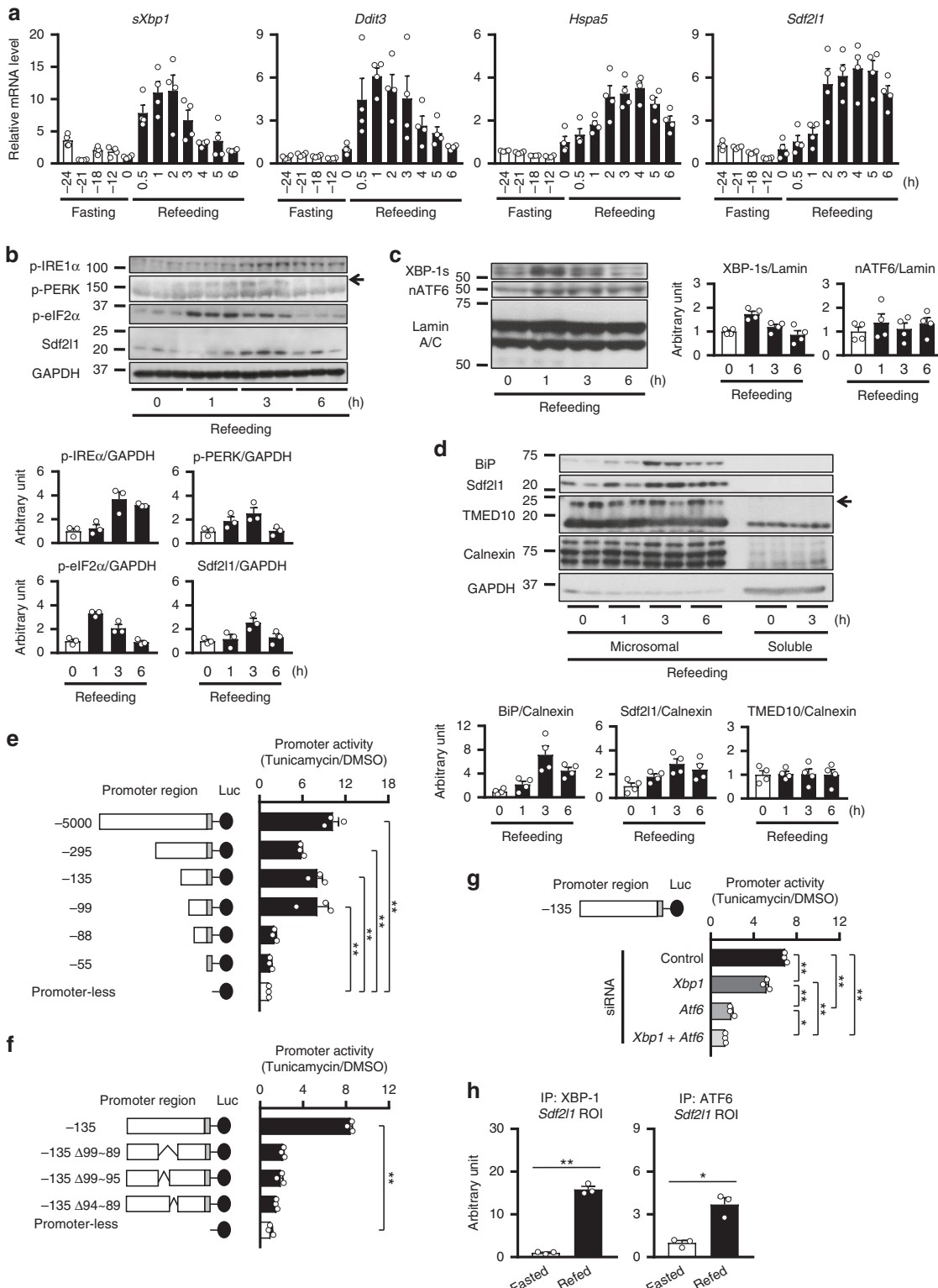

**Fig. 1** ER stress induced by feeding in the liver and regulation of Sdf2l1 expression. **a–d** C57BL/6J mice in an ad libitum-fed state were fasted for 24 h and then refed for 6 h, and ER stress markers were analyzed by **a** RT-PCR ($n = 4$), **b** Western blotting of total lysates ($n = 3$), **c** Western blotting of nuclear extracts (XBP-1s and nATF6 were detected at 55 and 50 kDa, respectively) ($n = 4$), and **d** Western blotting of microsomal fractions ($n = 4$). **e–g** Sdf2l1 promoter assays in Fao cells by transfecting luciferase (Luc) plasmids, with tunicamycin treatment, assessed with one-way ANOVA ($n = 3$), **e** using deletion mutants of upstream sequences, **f** using partial or total deletion mutants of the 11 bases of interest, and **g** using reporter plasmids, with knocking down of Xbp1 and/or Atf6. **h** ChIP assay, using antibodies recognizing XBP-1 or ATF6, for analysis of binding with the Sdf2l1 promoter in the liver, in a 24-h fasted state and a 3-h refed state ($n = 3$). The relative expression levels were normalized by the upstream region ($-3.5$ kb) and then by control IgG (rabbit or mouse IgG, respectively). Values of the data are expressed as mean ± SEM. *$P < 0.05$, **$P < 0.01$. Unpaired 2-tailed $t$-test was used for assessment

targeted by XBP-1s and nuclear ATF6 (nATF6)[10], which is conserved among mice, rats and humans (Supplementary Fig. 3e, f). Indeed, the activity was suppressed by knocking down of either transcription factor, in parallel with the efficiency of RNA interference, and importantly, it was almost blunted by knocking down of both (Fig. 1g and Supplementary Fig. 3g).

Moreover, chromatin immunoprecipitation (ChIP) assays showed that the binding of XBP-1s and nATF6 was elevated during refeeding with the region of interest (ROI) of the *Sdf2l1* promoter, as well as with the ERSE of the *Hspa5* promoter (Fig. 1h and Supplementary Fig. 3h). These data suggest that ER stress directly induces Sdf2l1 expression via XBP-1s and nATF6.

The next question was how Sdf2l1 functions in ER stress responses in the liver. We hypothesized that it could be involved in ER stress-associated degradation (ERAD), because Pmt1/2p, orthologs of Sdf2l1, enhance ubiquitination via *O*-mannosylation of unfolded proteins in yeast, as an initiation of ERAD[31–33]. We expressed the fusion protein Pmt2-Sdf2l1 in yeast lacking *Pmt2*, but failed to increase the mannosylation of a mutant secretory protein, a pro-region-deleted derivative of *Rhizopus niveus* aspartic proteinase I (Δpro), which was promoted by the restoration of Pmt2 (Supplementary Fig. 3i, j). Together with the fact that in mammals, *O*-mannosylation itself is scarcely detected with only a few exceptions[34], it is likely that Sdf2l1 does not have an *O*-mannosyltransferase activity. However, it still remained possible that Sdf2l1 could promote ERAD, and thus to test this possibility, we expressed Ins2[C96Y], a mutant insulin found in Akita mice, which is degraded by ERAD[35], in *Sdf2l1*-floxed murine embryonic fibroblasts (MEF) (Supplementary Fig. 4a–c). Accumulation of the ubiquitinated mutant insulin was markedly increased by the absence of Sdf2l1, which was reversed by reexpressing Sdf2l1 (Fig. 2a, b and Supplementary Fig. 4d). Interestingly, such accumulation was not observed by knocking down of BiP or thapsigargin-induced global dysfunction of the ER (Fig. 2c), although Sdf2l1 has been reported to bind to BiP in other tissues[26–28]. Moreover, ER stress was further enhanced by additional knocking down of *Hspa5* compared to knocking down of *Sdf2l1* alone (Fig. 2d).

The lack of enzyme activity and the functional difference with BiP suggested the existence of another partner of Sdf2l1. We explored it by mass spectrometric analysis of immunoprecipitated microsomal fraction of *Sdf2l1*-knockout MEF expressing Sdf2l1-FLAG (Supplementary Fig. 3i) and treated with tunicamycin. Among the 48 candidates, including DNAJB11 (also called ERdj3), a known partner of Sdf2l1 involved in protein folding[26,27,29], we screened ER-resident proteins and, of those tested, focused on transmembrane emp24-like trafficking protein 10 (TMED10) (Supplementary Table 2). TMED10 is the ortholog of p24, a membrane protein which interacts with Pmt1/2p and promotes ER export of unfolded proteins for ERAD in yeast[32], and is known to be involved in COPII vesicle-mediated protein transportation from the ER to the Golgi apparatus in mammals[36]. Although TMED10 showed almost constitutive expression patterns, contrary to Sdf2l1 (Fig. 1d and Supplementary Figs. 1a and 3a), Sdf2l1 bound to the 24 kDa isoform of TMED10, but not to βCOP, a key component of COPII vesicles, in the liver (Fig. 2e). Knocking down of *Tmed10* resulted in accumulated mutant insulin in primary hepatocytes, just as knocking down of *Sdf2l1* did (Fig. 2f and Supplementary Fig. 5a, b). In insulin-treated primary hepatocytes, which partially mimicked the liver in a fed state, knocking down of either *Sdf2l1* or *Tmed10* up-regulated ER stress markers. Interestingly, compared to knocking down of *Sdf2l1* alone, additional knocking down of *Hspa5* further up-regulated ER stress markers, while additional knocking down of *Tmed10* did not (Fig. 2g and Supplementary Fig. 5a, b). These data imply that Sdf2l1 plays a key role in the process of ERAD, in

cooperation with TMED10 and in an independent manner of BiP. Moreover, it is known that Pmt1/2p promotes Hrd1p-mediated ERAD in yeast[32], and actually also in mice, the effects of knocking down of *Sdf2l1* were mainly dependent on Hrd1 (Supplementary Fig. 5c, d), the major E3 ligase in the canonical pathway of ERAD in mammals[37].

**Sdf2l1 modulates ER stress responses and metabolism**. We then assessed the physiological and pathophysiological roles of Sdf2l1 in vivo. We knocked down *Sdf2l1* specifically in the liver of wild-type mice by adenovirus-mediated gene transfer of shRNA for *Sdf2l1* (Fig. 3a, b). Although these mice showed no changes in body weight, ER stress was enhanced during refeeding, accompanied by up-regulation of genes involved in oxidative stress and inflammation (Fig. 3a, c and Supplementary Fig. 6a, b).

With regards to glucose metabolism, the knocking down of *Sdf2l1* resulted in elevated plasma glucose levels in an ad libitum-fed state, systemic insulin resistance, glucose intolerance, and impaired insulin signaling at the level of Akt after refeeding compared to findings for the mice treated with the control virus (Fig. 3c–f and Supplementary Fig. 6c). Hyperinsulinemic-euglycemic clamp studies also showed systemic insulin resistance, enhanced gluconeogenesis, accompanied by impaired insulin signaling and elevated gluconeogenic gene expression in the liver (Fig. 3g–i), although the impairment was not evident in skeletal muscle (Fig. 3g and Supplementary Fig. 6d).

We examined lipid metabolism as well, and found that triglyceride contents were markedly increased in the liver by knocking down of *Sdf2l1* (Fig. 3j, k). Indeed, expression of genes involved in fatty acid synthesis and nuclear translocation of the related transcription factors was significantly increased (Fig. 3a and Supplementary Fig. 6b), despite up-regulated lipolytic gene expression in the liver, probably in a compensatory manner, and non-altered plasma free fatty acid levels (Supplementary Fig. 6b, e).

We also generated liver-specific *Sdf2l1*-knockout mice by administering adenovirus expressing Cre recombinase to *Sdf2l1*-floxed mice (Fig. 4a and Supplementary Fig. 4a–c), which exhibited similar phenotypes for a longer period (Fig. 4b–i), further supporting data of the knocking down model.

**Impaired ER stress responses in obesity and diabetes**. We then investigated the role of Sdf2l1 in the development of diabetes and fatty liver disease. To this end, we first examined the profile of the ER responses during fasting and feeding in *db/db* mice, as a model of obesity, diabetes, and fatty liver disease. We found that ER stress sensors were activated in parallel with enhanced protein synthesis markers (Fig. 5a and Supplementary Fig. 7a, b). As for the IRE1α-ATF6 branch, *sXbp1* mRNA was up-regulated only slightly (Fig. 5b), possibly suggesting impaired and delayed splicing activity of IRE1α. XBP-1s and nATF6 located in the nucleus were significantly decreased early in the refed state (Fig. 5c), as we and others have previously reported[19,20,38]. ChIP assay showed that the binding to the *Sdf2l1* promoter, as well as to the *Hspa5* promoter, was suppressed markedly in case of XBP-1, and to a much less extent in case of ATF6 (Fig. 5d and Supplementary Fig. 7c). Consequently, the downstream chaperones were markedly down-regulated, although TMED10 protein showed little change in expression (Fig. 5b, e). Regarding the PERK branch, eIF2α was less activated, possibly due to dysfunction of PERK as a kinase, and *Ddit3* mRNA was down-regulated as well (Fig. 5a, b and Supplementary Fig. 7b).

These data suggest the existence of a vicious cycle; despite the enhanced ER stress, downstream molecules of the cascade that are expected to cope with ER stress could be suppressed in

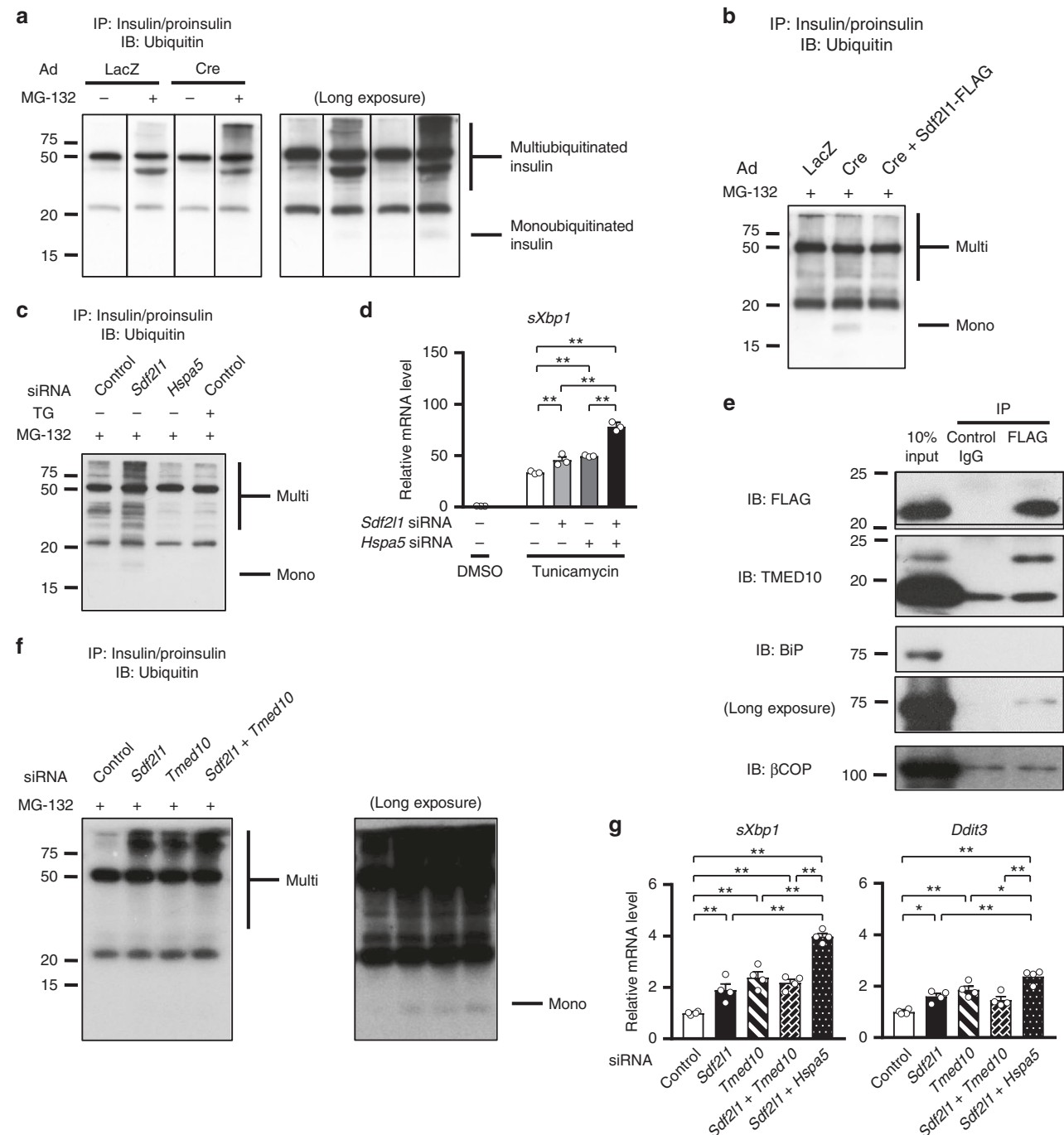

**Fig. 2** Functions of Sdf2l1 in vitro. **a–c**, **f** Cultured cells were infected with Ad-Ins^C96Y, whose lysates were immunoprecipitated for Western blotting: **a**, **b** Sdf2l1-floxed MEF cells, with **a** co-infection of Ad-Cre to knock out Sdf2l1, and with **b** further co-infection of Ad-Sdf2l1-FLAG to restore the expression; **c** NIH/3T3 cells, with either Sdf2l1 or Hspa5 knocked down, or with treatment with thapsigargin for 6 h; **f** primary hepatocytes, with Sdf2l1 or Tmed10 knocked down. In **a**, the lanes were run on the same gel but were noncontiguous. Multi: multiubiquitinated insulin, Mono: monoubiquitinated insulin. **d** RT-PCR to analyze ER stress marker gene expression in NIH/3T3 cells with Sdf2l1 and/or Hspa5 knocked down, treated with tunicamycin treatment ($n = 3$). **e** Wild-type mice were administered with Ad-Sdf2l1-FLAG intravenously ($2.0 \times 10^7$ PFU/g body weight [BW]), and refed for 6 h after fasting for 24 h, whose microsomal fractions were immunoprecipitated for Western blotting. **g** RT-PCR to analyze ER stress marker gene expression in primary hepatocytes, with ER-resident molecule(s) knocked down, treated with 100 nM insulin for 2 h after serum starvation for 16 h ($n = 4$). Values of the data are expressed as mean ± SEM. *$P < 0.05$, **$P < 0.01$. One-way ANOVA was used for assessment

expression or insufficiently activated, which we call ER stress response failure, resulting in further excessive ER stress. Among them, Sdf2l1 was most prominently down-regulated both during fasting and feeding, accompanied by delayed nuclear localization of XBP-1s during refeeding, presumably due to the decreased

insulin action to promote the translocation of XBP-1s by binding to p85[20,38].

ER stress marker genes were similarly down-regulated in another model of severe insulin resistance, ob/ob mice (Supplementary Fig. 7d). In diet-induced obesity, physiological regulation

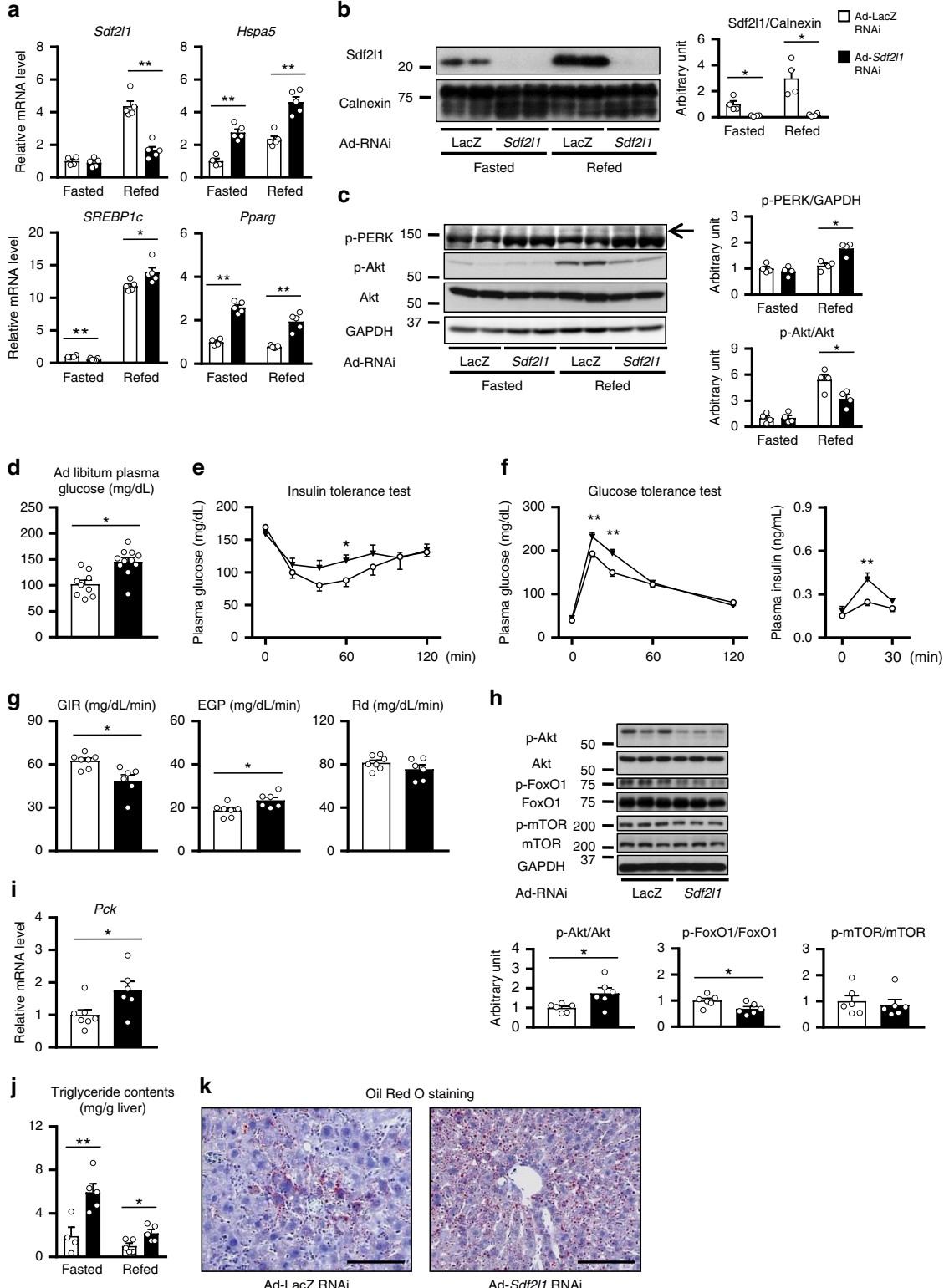

**Fig. 3** Functions of Sdf2l1 in vivo. Wild-type mice were administered with Ad-*Sdf2l1* RNAi intravenously (2.0 × 10⁷ PFU/g BW) to knock down the gene in the liver. **a–c** ER stress and insulin signaling in the liver ($n = 4$–5), analyzed by **a** RT-PCR ($n = 4$–5), **b** Western blotting of microsomal fractions ($n = 4$), and **c** Western blotting of total lysates ($n = 4$). **d–f** Effects of knocking down on glucose metabolism ($n = 9$–11): **d** ad libitum-fed plasma glucose, **e** plasma glucose in insulin tolerance test (ITT), after intraperitoneal injection of human regular insulin (0.75 U/kg BW), **f** plasma glucose and insulin levels in an oral glucose tolerance test (OGTT), after oral administration of glucose (0.75 g/kg BW), following 16 h of fasting. **g–i** Results of hyperinsulinemic-euglycemic clamp studies after 3 h of fasting (2.5 mU/kg/min, $n = 6$–7), in wild-type mice with Sdf2l1 knocked down: **g** glucose infusion rate (GIR), endogenous glucose production (EGP), and rate of glucose disappearance (Rd); **h, i** the liver samples after the clamp studies were analyzed by **h** Western blotting and **i** RT-PCR. **j, k** Effects of knocking down on lipid metabolism, analyzed with **j** triglyceride contents quantification ($n = 4$–5), and **k** Oil Red O staining. Values of the data are expressed as mean ± SEM. *$P < 0.05$, **$P < 0.01$. Unpaired 2-tailed $t$-test was used for assessment. Scale bars: 100 μm

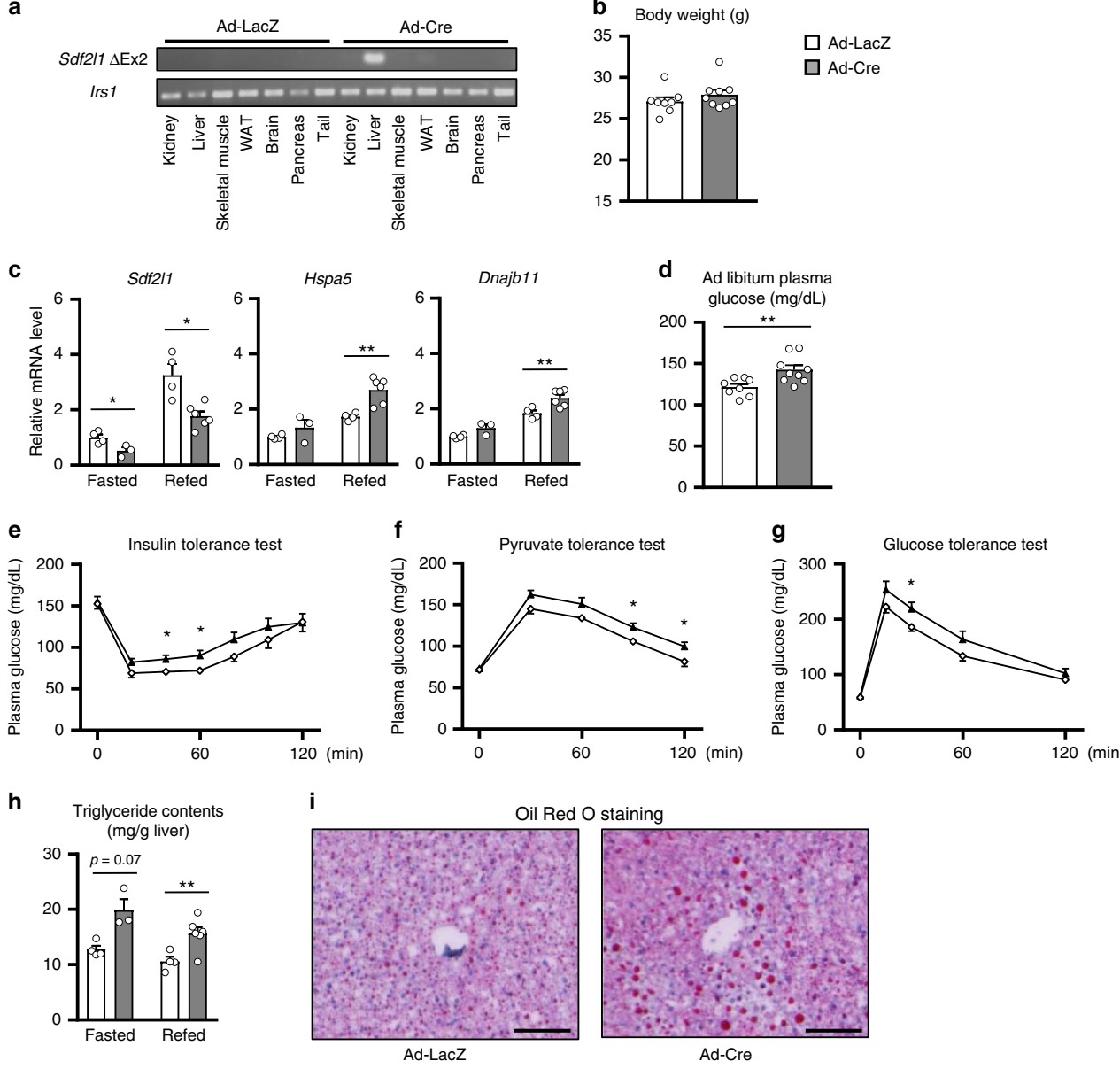

**Fig. 4** Generation and phenotypes of the knock out model of *Sdf2l1*. *Sdf2l1*-floxed mice were administered with Ad-Cre (3.0 × 10$^7$ PFU/g BW), to generate a liver-specific *Sdf2l1* knockout model in adults. **a** Detection of deleted exon 2 of the *Sdf2l1* gene by PCR. **b**, **d–i** Metabolic phenotypes analyzed 3–6 weeks after the adenovirus administration (*n* = 8–9): **b** body weight, **d** ad libitum-fed plasma glucose, **e** plasma glucose in insulin tolerance test (ITT), after intraperitoneal injection of human regular insulin (0.75 U/kg BW), **f** plasma glucose levels in an pyruvate tolerance test, after intraperitoneal injection of pyruvate (1.5 g/kg BW), **g** plasma glucose levels in an oral glucose tolerance test (OGTT), after oral administration of glucose (0.75 g/kg BW), following 16 h of fasting. **c** ER stress in the liver (*n* = 3–6), analyzed by RT-PCR. **h**, **i** Effects on lipid metabolism, analyzed with **h** triglyceride contents quantification (*n* = 3–6), and **i** Oil Red O staining. Values of the data are expressed as mean ± SEM. *\*P* < 0.05, *\*\*P* < 0.01. Unpaired 2-tailed *t*-test was used for assessment. Scale bars: 100 μm

of ER stress responses was partially impaired in parallel with mild insulin resistance (Supplementary Fig. 7e).

**Effects of restoring hepatic expression of Sdf2l1.** In order to rescue ER stress response failure in *db/db* mice, we first enhanced hepatic expression of *sXbp1*, an upstream transcription factor, by adenovirus-mediated gene transfer. It failed, however, to show full recovery of expression of downstream chaperones including *Sdf2l1* and consequently insulin resistance (Supplementary Fig. 8a–c), which urged us to restore the suppressed expression of Sdf2l1 (Fig. 6a, b). It did not affect body weight, but did suppress activation of PERK, and lowered ad libitum plasma glucose levels (Fig. 6c, d and Supplementary Fig. 8d). Glucose tolerance was

improved, accompanied by partial recovery of early phase insulin secretion after glucose challenge, phosphorylation of Akt was enhanced after feeding, and hyperinsulinemic-euglycemic clamp studies showed improvement in systemic insulin resistance and hepatic glucose production (Fig. 6c, e, f). Triglyceride contents were decreased (Fig. 6g, h), accompanied by suppressed gene expression involved in fatty acid synthesis (Supplementary Fig. 8e).

In accordance with the findings in vitro (Fig. 2d, g), compared to single restoration of Sdf2l1, additional restoration of BiP, had the larger beneficial effects (Fig. 6i, j and Supplementary Fig. 8f–j),

suggesting that Sdf2l1 could improve insulin sensitivity independently of BiP, at least in part.

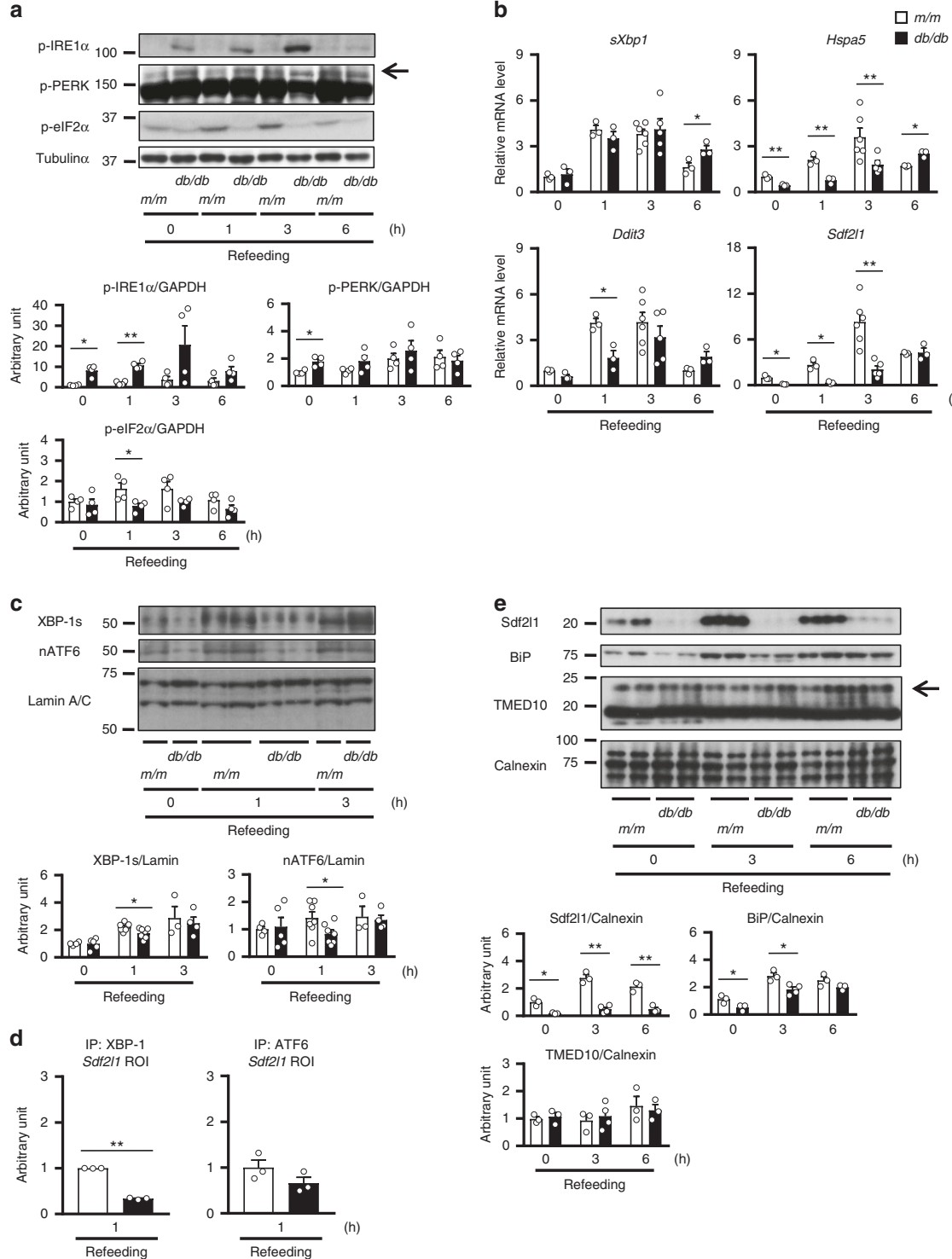

**Fig. 5** ER stress responses in obesity and diabetes. Both *db/db* mice and *m/m* mice were fasted for 24 h and refed for 6 h. **a**–**c**, **e** ER stress markers were analyzed by **a** Western blotting of total lysates ($n = 4$), **b** RT-PCR ($n = 3$–6), **c** Western blotting of nuclear extracts ($n = 3$–7), and **e** Western blotting of microsomal fractions ($n = 3$–4). **d** ChIP assay, using antibodies recognizing XBP-1 or ATF6, for analysis of binding of transcription factors with the *Sdf2l1* promoter in the liver, in a 1-h refed state after 24 h of fasting ($n = 3$). The relative expression levels were normalized by the upstream region (−3.5 kb) and then by control IgG (rabbit or mouse IgG, respectively). Values of the data are expressed as mean ± SEM. \*$P < 0.05$, \*\*$P < 0.01$. Unpaired 2-tailed *t*-test was used for assessment

**Impaired ER stress responses in diseases in humans**. Lastly, we assessed whether impaired ER stress responses could be associated with progression of human diseases by examining data from 64 male subjects with suspected nonalcoholic fatty liver disease (NAFLD) who underwent liver biopsy after oral glucose tolerance test (OGTT) early in the morning, partially mimicking the fed state (Table 1). We examined gene expression of *sXBP1*, *SDF2L1*, and *HSPA5*. We also evaluated the *SDF2L1/sXBP1* ratio

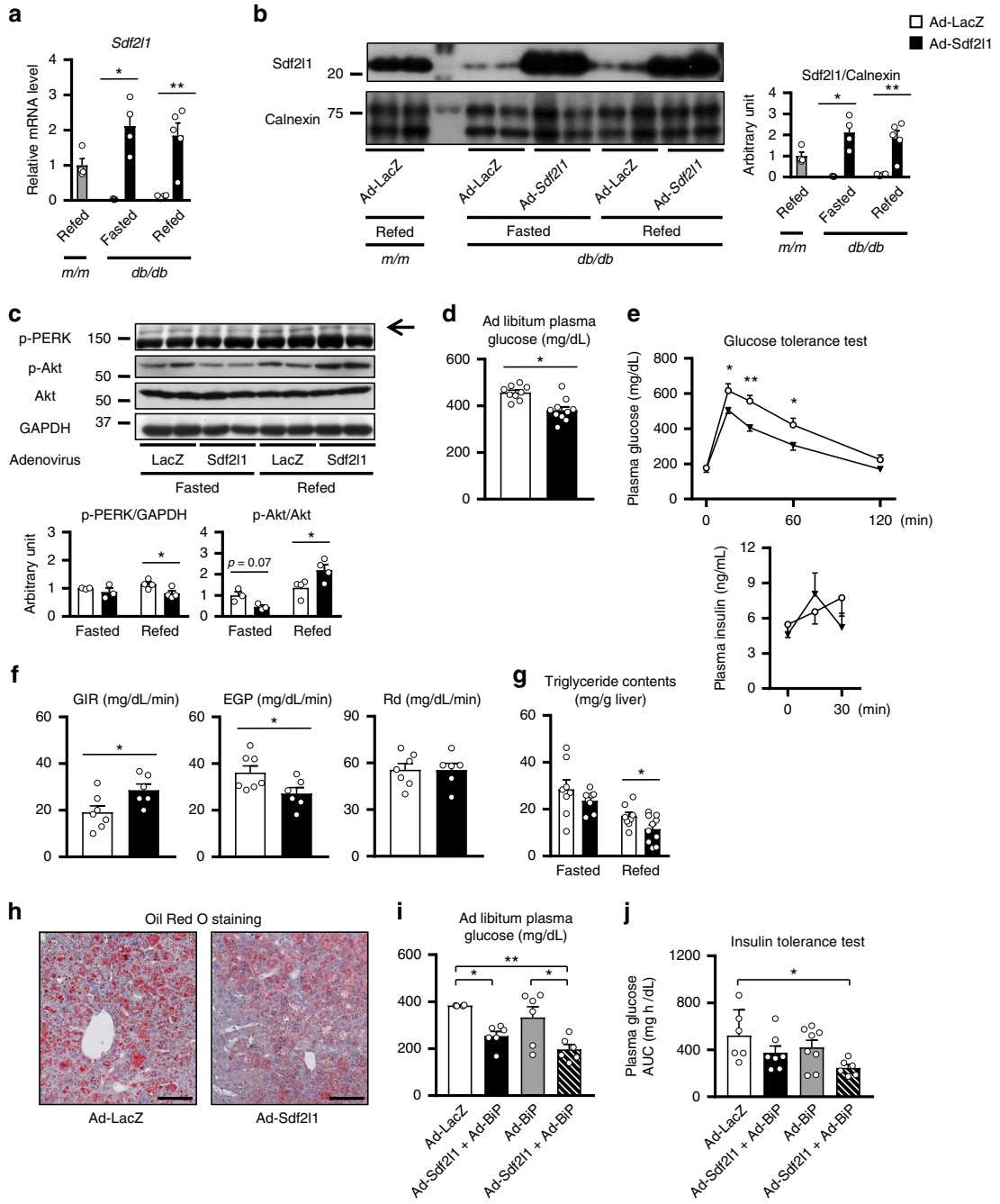

**Fig. 6** Restoration of Sdf2l1 expression in an obesity and diabetes model. **a–h** *db/db* mice were administered with Ad-Sdf2l1 intravenously ($2.5 \times 10^7$ PFU/g BW), for restoration of the gene expression in the liver. **a–c** ER stress and insulin signaling in the liver, analyzed by **a** RT-PCR ($n = 3$-5), **b** Western blotting of microsomal fractions ($n = 3$-5), and **c** Western blotting of total lysates ($n = 3$-4). **d–f** Effects of the restoration on glucose metabolism ($n = 9$-10), including **d** plasma glucose levels in an ad libitum-fed state, **e** plasma glucose levels in OGTT, as well as insulin levels, after oral administration of glucose (0.75 g/kg BW), following 24 h of fasting. **f** Results of hyperinsulinemic-euglycemic clamp studies (12.5 mU/min/kg, $n = 6$-7), after 3 h of fasting. **g**, **h** Effect of the restoration on lipid metabolism, analyzed with **g** triglyceride contents quantification ($n = 8$-10), and **h** Oil Red O staining. **i**, **j** Phenotypes of *db/db* mice co-administered with Ad-Sdf2l1 and Ad-BiP-HA, assessed with one-way ANOVA: **i** plasma glucose in an ad libitum-fed state ($n = 4$-6), and **j** the area under the curve (AUC) of plasma glucose in ITT, after intraperitoneal injection of human regular insulin (1.5 U/kg BW) ($n = 6$-8). Values of the data are expressed as mean ± SEM. *$P < 0.05$, **$P < 0.01$. **a–g** Unpaired 2-tailed *t*-test was used, and **i**, **j** one-way ANOVA was used for assessment. Scale bars: 100 µm

and the *HSPA5/sXBP1* ratio, because the lower ratios could be attributed to attenuated nuclear localization of XBP-1s, possibly due to impaired insulin signaling, resulting in ER stress response failure (Fig. 7a).

It was first revealed that expression of *SDF2L1* was negatively correlated with glycemic control (Fig. 7b), and we focused on the presence or absence of diabetes thereafter. Although it was natural that the diabetic subjects showed higher HbA1c and

**Table 1 Background characteristics of the human subjects**

|  | All ($n = 64$) | Nondiabetic, pre-match ($n = 39$) | Nondiabetic, post-match ($n = 25$) | Diabetic ($n = 25$) |
|---|---|---|---|---|
| Age (year) | 51.5 ± 1.8 | 47.2 ± 2.3 | 52.6 ± 2.4 | 58.1 ± 2.5 |
| Body mass index (kg/m$^2$) | 28.8 ± 0.5 | 28.8 ± 0.8 | 29.4 ± 0.8 | 28.8 ± 0.6 |
| Comorbid diabetes (No. [%]) | 25 (39.0%) | – | – | – |
| HbA1c (%) | 6.08 ± 0.10 | 5.62 ± 0.05 | 5.67 ± 0.08 | 6.79 ± 0.15 |
| HOMA-R | 3.65 ± 0.30 | 2.89 ± 0.22 | 3.28 ± 0.25 | 4.82 ± 0.64 |
| Ethanol intake (g/day) | 11.0 ± 1.6 | 11.4 ± 1.8 | 10.0 ± 2.4 | 14.2 ± 2.9 |
| ALT (IU/L) | 41.8 ± 2.8 | 40.0 ± 3.3 | 43.1 ± 4.4 | 44.5 ± 5.1 |
| AST (IU/L) | 68.6 ± 5.8 | 69.3 ± 6.6 | 66.4 ± 6.6 | 67.5 ± 10.9 |
| NAS | 3.3 ± 0.2 | 3.2 ± 0.3 | 3.7 ± 0.4 | 3.6 ± 0.4 |
| Stage | 1.6 ± 0.2 | 1.3 ± 0.2 | 1.6 ± 0.2 | 2.0 ± 0.2 |

The list to show background characteristics of the human subjects analyzed. Among the 39 nondiabetic patients, 25 were selected as those matched with the 25 diabetic patients based on age, NAS, and stage
Plus–minus values are mean ± SEM
*HOMA-R* homeostasis model assessment for insulin resistance, *ALT* alanine transaminase, *AST* aspartate transaminase, *NAS* NAFLD activity score

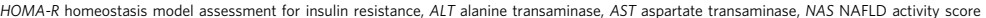

**Fig. 7** ER stress responses in human subjects. Expression of ER stress marker genes and the ratios between them, in liver biopsy samples of human male subjects with suspected nonalcoholic fatty liver disease (NAFLD), analyzed by RT-PCR. **a** Schematic description of factors affecting expression of marker genes and ratios. **b** Correlation with glucose tolerance ($n = 64$). **c** Comparison between diabetic ($n = 25$) and matched nondiabetic subjects ($n = 25$). **d**–**h** Correlation with progression of comorbid diseases in diabetic ($n = 25$) and matched nondiabetic subjects ($n = 25$), including **d**–**f** insulin resistance, and **g**, **h** staging of NASH. Data are shown in scatter dot plot with the median and interquartile range. *$P < 0.05$, **$P < 0.01$, T tertile. **b**, **g**, **h** Spearman's rank correlation was used, and **c**, **e**, **f** the Mann–Whitney $U$ test was used for assessment

HOMA-R, they were older and showed advanced stage of NASH. Therefore, among the 39 nondiabetic subjects, we selected 25 as those matched with the 25 diabetic subjects based on age and histological findings, i.e., NAS and stage (Table 1). Expression of *SDF2L1* was significantly down-regulated in those with diabetes (Fig. 7c), whereas expression of the other genes was not (Supplementary Fig. 9). Moreover, in insulin resistant subjects with diabetes, despite elevated expression of *sXBP1*, the *SDF2L1/sXBP1* ratio and the *HSPA5/sXBP1* ratio were significantly lower (Fig. 7d–f). Similarly, in those with diabetes, *sXBP1* was positively, but the *SDF2L1/sXBP1* ratio was negatively, correlated with stage of NASH (Fig. 7g, h). These data suggest that, similar to our findings in murine models, in patients with diabetes, impaired responses to ER stress, as well as enhanced ER stress, could be associated with the progression of insulin resistance and steatohepatitis.

## Discussion

ER stress has been thought to cause metabolic disorders when induced chronically in various tissues[39]. Here we show, however, that a physiological stimulus of feeding induces transient ER stress responses (just for a few hours) in the liver.

During feeding in mice, *Sdf2l1* shows a marked induction in expression among ER stress-related genes. It is regulated by XBP-1s and nATF6, which are known to regulate chaperones, and Sdf2l1 appears to terminate ER stress, as chaperons do. Given that Sdf2l1 has an ER-retention-like motif, it is expected to traffic between the ER and the Golgi apparatus and cope with ER stress, even during feeding, although Sdf2l1 is not likely to be a component of COPII vesicles. Sdf2l1 exerts its effect mainly through the mechanism regulating ERAD of, probably, unfolded proteins in general, and we identify a counterpart, TMED10, a membrane protein involved in protein trafficking. Sdf2l1 is known to bind with BiP[26–28], and indeed weak interaction between them was detected in the liver, but TMED10 seems the major counterpart of Sdf2l1, in terms of both interaction and function. Moreover, it is suggested that Sdf2l1 could be involved in shuttling of substrates for ERAD ubiquitinated mainly by Hrd1 to the proteasome outside the ER, just as Dnajb2, which works as a chaperone inside the ER and is also involved in the shuttling outside the ER[37]. In yeast, Pmt1/2p, the orthologs of Sdf2l1, are anchored in the ER membrane and interact with p24, the ortholog of TMED10. Sdf2l1 in mammals, however, lacks a transmembrane domain, and thus it is probable that it requires TMED10 as a counterpart serving as a platform in order to work near or even through the ER membrane.

We have found that in the liver, impaired induction of Sdf2l1 results in sustained ER stress, leading to insulin resistance and increased triglyceride contents, even with a normal-chow diet, indicating that dysregulation of ER stress by suppression of Sdf2l1 is a causal factor of metabolic disorders. Together with the previous reports showing that ablation of key molecules in ER stress responses links impaired glucose and lipid metabolism in mice fed on a high-fat diet[11,17,40], our data strongly suggest that an appropriate transient response to ER stress is induced physiologically during feeding and terminated by Sdf2l1, and that this process may be important for the maintenance of nutrient homeostasis.

Importantly, the downstream molecules involved in ER stress responses fail to be fully induced or activated in obesity and diabetes, despite activation of the upstream ER stress sensors, reflecting enhanced ER stress. We call such a discrepancy, ER stress response failure. On the other hand, when the ER stress sensors are not fully activated for some reasons, the downstream ER stress responses are not sufficiently induced, even under

enhanced ER stress[41]. This condition might be called ER stress sensing failure.

Our data reveal that a vicious cycle exists, with enhanced ER stress and ER stress response failure, in the liver of diabetes-associated insulin resistance and NAFLD/NASH in mice and humans. In the 'two-hit theory', ER stress is thought to be one of the second hits in the progression of NASH, and our data suggest that this appears to be the results of ER stress response failure. Insulin resistance in the first hit can cause dysfunction of XBP-1s, presumably due to delayed insulin-mediated translocation to the nucleus[20,38], resulting in the suppression of the termination signal for ER stress such as induction of SDF2L1, leading to sustained ER stress that can function as a second hit. Indeed, in a diabetic condition, where insulin resistance fails to be compensated and thus insulin action is insufficient, there is a discrepancy between the induction of XBP-1s paralleled with ER stress and chaperon production, including SDF2L1, reflected by both ER stress and insulin action. Therefore, the lower *SDF2L1/sXBP1* ratio could be a much better biomarker than other ER stress-related genes, such as *sXBP1* alone, to reflect not only ER stress but also ER stress response failure that leads to progression of diabetes-associated diseases in the liver. It should be investigated in future works whether such ER stress response failure, observed in male mice and male subjects, could contribute to the development of NASH in female subjects as well, which is largely affected by menopause.

These data suggest that dys-regulation of Sdf2l1 is a missing link between insulin resistance and NAFLD/NASH, and can serve as a therapeutic target for diabetes and NAFLD/NASH. Indeed, it is known that insulin sensitizers can ameliorate hyperglycemia, as well as NASH[42], although insulin signaling promotes anabolism of lipids. Based on our data, this may be partly through efficient induction of Sdf2l1 and BiP by XBP-1s, while over-expression of XBP-1s alone is not enough, because the translocation to the nucleus remains impaired without improving insulin resistance. It remains to be clarified whether the beneficial effects of Sdf2l1 restoration could be sustained for a longer period, for weeks or months. Besides, to develop more effective drugs for these diseases by shutting down the vicious cycle, it may be useful to identify the effectors downstream of Sdf2l1 to regulate ERAD, besides the strategy to promote the translocation of XBP-1s to the nucleus to up-regulate Sdf2l1[20,38].

In this context, lipid species might be also important. For example, ceramides are accumulated by ER stress, which in turn induces insulin resistance[39,43]. It is also likely that lipid species derived from feeding could affect those constituting the ER membrane, which in turn could affect functions of the ER in the liver, namely lipogenesis and protein synthesis. Thus, it remains to be investigated how lipid species in the liver, especially those isolated in microsomal fractions, are affected in response to feeding or chemically induced ER stress, and how such responses are disturbed by dys-regulation of Sdf2l1 or obesity-induced diabetes.

Overall, feeding induces physiological and transient ER stress in the liver, and induced Sdf2l1 appropriately terminates ER stress, in cooperation with TMED10, and contributes to normal glucose and lipid metabolism. In obesity and diabetes, impaired ER stress termination signals, including the down-regulation of Sdf2l1 that is caused by decreased insulin signaling, sustains ER stress and exacerbates insulin resistance, creating a vicious cycle (Supplementary Fig. 10). Thus, Sdf2l1 is expected to be a therapeutic target and a sensitive biomarker in obesity-associated diseases.

## Methods

**Generation of the mutant mice**. *Sdf2l1*-floxed mice (Accession No. CDB0801K [http://www2.clst.riken.jp/arg/mutant%20mice%20list.html]) were generated as described elsewhere (http://www2.clst.riken.jp/arg/Methods.html). To generate the

targeting vector, a genomic fragment of the *Sdf2l1* locus was obtained from the RP23-153F8 BAC clone (BACPAC Resources), and a 472 bp-region containing exon 2 of the *Sdf2l1* gene was flanked by *loxP* sites (Supplementary Fig. 4a). Targeted ES clones were microinjected into ICR 8-cell stage embryos, and injected embryos were transferred into pseudopregnant ICR females. The resulting chimeras were bred with C57BL/6 J mice, and then with *ACTB-FLPe* mice (Stock Number 005703; Jackson Laboratory) to eliminate the neomycin resistance (*Neo*) gene flanked by *FRT* sites. Experiments were performed using *Sdf2l1*-floxed (*ΔNeo*) homozygous mice, after *Neo* elimination and backcrossing with C57BL/6 J mice 4 times. Southern blot analysis was performed by separation of genomic DNA by electrophoresis on a 0.6% agarose gel, followed by transfer onto Hybond-XL membranes (GE Healthcare) before hybridization with the 32P-random-prime-labeled probe[44]. The 758bp-5′ external probe amplified with genomic murine DNA as the template with the primers described later (Supplementary Fig. 4a). Geno-typing PCR was performed using ExTaq (TaKaRa) and the primers described later (Supplementary Fig. 4b). Uncropped images are shown in Supplementary Fig. 11.

**Mice**. C57BL/6 J mice and B6.Cg-*Lep^{ob}*/J (*ob/ob*) mice were purchased from Oriental Yeast, and BKS.Cg- + *Lepr^{db}*/ + *Lepr^{db}*/Jcl (*db/db*) mice from CLEA Japan, and 8-week-old to 10-week-old male mice were subjected to experiments. All mice were housed under a 12-h light/12-h dark cycle and had free access to sterile water and the following pellet food: Oriental MF diet (Oriental Yeast), consisting of 23.6% (v/v) protein, 5.3% fat, 54.4% nitrogen free extract, 6.1% ash, 2.9% fiber, and 7.7% water. In fasting and refeeding experiments, mice were fed with pellet food for 6 h, unless otherwise indicated, after 24-h fasting in individual cages[6,8,18]. Experiments in an ad libitum-fed state were performed immediately after the beginning of the light cycle. In nutrient deprivation experiments, mice were fed with mixed chow, consisting of casein, corn oil, and corn starch (Oriental Yeast) at the ratio of 23.6:5.3:54.4; protein-free chow consisting of corn oil and corn starch at the ratio of 5.3:54.4; or lipid-free chow consisting of casein and corn starch at the ratio of 23.6:54.4. To generate a diet-induced obesity model, we fed mice with High Fat Diet 32 (CLEA Japan), consisting of 25.5% protein, 32.0% fat, 29.4% nitrogen free extract, 4.0% ash, 2.9% fiber, and 6.2% water. In fasting and refeeding experiments using this model, mice, including those on normal chow, were fed with high-fat diet. Body weight and blood glucose levels were matched among groups in experiments allocating mice to different interventions. The animal care and experimental procedures were approved by the Animal Care Committee of Graduate School of Medicine, the University of Tokyo. All relevant ethical guidelines were followed.

**Metabolic studies**. For insulin tolerance tests (ITTs) or pyruvate tolerance tests (PTTs), mice received intraperitoneal administration of human insulin (Humalin R; Eli Lilly) in an ad libitum-fed state, or pyruvate sodium after an overnight fast, respectively. For OGTTs, mice received oral administration of glucose after an overnight fast. Blood glucose levels were measured using a Glutest sensor (Sanwa Chemical) at the indicated time points, the plasma insulin levels were measured using an ELISA kit (Morinaga), and the plasma free fatty acid levels were measured using a quantification kit (WAKO). In hyperinsulinemic-euglycemic clamp studies, human insulin was continuously infused, and the blood glucose concentration was maintained at approximately 120 mg/dL by infusing glucose, containing [6,6-2H2] glucose (Sigma). At the end of the clamp studies, a bolus of 2-deox y-[3 H]glucose (2-[3 H]DG) was administered for estimation of glucose uptake[45]. The investigators were blinded to the genotype of mice or intervention to which mice were subjected when assessing metabolic phenotypes.

**Cell culture**. HEK293 cells (ECACC 85120602) were cultured in DMEM (Sigma) supplemented with 10% FBS (Biowest), Fao cells (ECACC 89042701) in RPMI 1640 (GIBCO) supplemented with 10% FBS, and NIN/3T3 cells (ECACC 93061524) in DMEM supplemented with 10% calf serum (Biowest). Embryonic fibroblasts were isolated from 13.5-dpc homozygous embryos by treatment with trypsin (GIBCO), and resuspended and cultured in DMEM supplemented with 10% FBS, 1 mM sodium pyruvate (GIBCO), and MEM Non-Essential Amino Acids Solution (GIBCO)[46]. To isolate primary hepatocytes, the liver was perfused by cannulation in the portal vein with HBSS containing collagenase (Nitta Gelatin) and dispase (WAKO), and the isolated cells were resuspended and cultured in WME (GIBCO) supplemented with 10% FBS, 100 nM dexamethasone, and 1 nM human insulin[47].

**Chemical treatment**. Tunicamycin, thapsigargin (both from Sigma), and MG-132 (Merck Millipore) were diluted in dimethyl sulfoxide (DMSO) and added to the medium of cultured cells, at the concentration of 1 μg/mL, 1 μM, and 10 μM, respectively. Streptozotocin (Sigma) was diluted in citrate sodium buffer, and administered intraperitoneally at the dose of 150 mg/kg of body weight (BW) twice with an interval of 4 days. Mice were used for experiments 2 weeks after the treatments.

**Microarray analysis**. The data set was obtained from the GEO (https://www.ncbi.nlm.nih.gov/geo/) and analyzed using GEO2R on line.

**Gene expression analysis**. RNeasy Mini Kit (QIAGEN) was used to prepare total RNA from mouse tissues and cultured cells, and ReliaPrep RNA Tissue Miniprep System (Promega) from human samples. Reverse-transcription reaction was carried out with a High Capacity cDNA Reverse Transcription Kit (Applied Biosystems), after treatment with DNAse (Promega). Quantitative PCR were performed using ABI Prism 7900, with TaqMan Gene Expression Master Mix or Power SYBR Green PCR Master Mix (Applied Biosystems). The relative expression levels were normalized by the expression level of *Ppia* (encoding Cyclophilin A) in samples of mouse tissues and cultured cells, control IgG and control upstream region in ChIP assays, and *GAPDH* in human samples, respectively. TaqMan Assays were purchased from Applied Biosystems: *Hspa5* (BiP), Mm00517691_m1; *Ddit3* (CHOP), Mm00492097_m1; *Sdf2l1*, Mm00452079_m1; *Ero1lb*, Mm00470754_m1; *Pdia3*, Mm00433130_m1; *Sdf2*, Mm00485985_m1; *Pparg*, Mm00440945_m1; *Pten*, Mm00477208_m1; *Sod1*, Mm01344233_g1; *Sod2*, Mm01313000_m1; *Tnf*, Mm99999068_m1; *Ppara*: Mm00627559_m1; *Fasn*, Mm00662319_m1; *Acacb*, Mm01204671_m1; *Pck1*: Mm00440636_m1.

**Primers**. Primers were designed in Primer3 (http://bioinfo.ut.ee/primer3-0.4.0/), unless otherwise described, and synthesized by Invitorgen, whereas TaqMan Probes were synthesized by Applied Biosystems (see Table 2).

**Antibodies**. Anti-GRP78/BiP (H-129, 1:250 for immunoblotting), anti-PERK (B-5, for immunoprecipitation), anti-XBP-1 (M-186, 1:250), anti-ubiquitin (P4D1, 1:200), anti-Hrd1 (A-21, 1:200), and anti-IRβ (C-19, 1:1000) antibodies were purchased from Santa Cruz; anti-ATF6 (70B1413.1, 1:500) from IMGENEX; anti-HA (12CA5, 1:10000) from Roch; anti-calnexin (PM060, 1:1000) from MBL; anti-insulin/proinsulin (ab8304, for immunoprecipitation), and anti-βCOP (ab2899, 1:1000) from Abcam; anti-FLAG (M2, 1:2500) from Sigma; anti-tubulinα (RB-9281, 1:1000) from Thermo Scientific; anti-pY (4G10, 1:1000), anti-IRS-1 (32682, 1:1000), and anti-p85 (27891, 1:4000) from Upstate; and the others, anti-p-PERK (3719, 1:1000), anti-PERK (3192, 1:1000 for immunoblotting), anti-p-eIF2α (3597, 1:1000), anti-eIF2α (9722, 1:500), anti-LaminA/C (2032, 1:1000), anti-p-S6K (9204, 1:1000), anti-p-4E-BP1 (9459, 1:800), anti-p-Akt (9271, 1:1000), anti-Akt (9272, 1:1000), anti-p-FoxO1 (9461, 1:1000), anti-FoxO1 (9454, 1:1000), anti-p-mTOR (2971, 1:1000), anti-mTOR (2972, 1:2000), and anti-GAPDH (2118, 1:5000) antibodies were from Cell Signaling. Anti-Sdf2l1 antibody (1:750) was generated by Immuno-Biological Laboratories, by immunizing rabbits with recombinant murine Sdf2l1, as described later. Anti-p-IRE1α antibody (1:1000) and anti-IRE1α antibody (1:1000) were kind gifts from Dr. Fumihiko Urano (Washington University in St. Louis). Anti-TMED10 antibody (1:1000)[54] was a kind gift from Prof. Masahiro Hosaka (Akita Prefectural University) and Prof. Tetsuro Izumi (Gunma University). Anti-RNAP I antibody (1:10000) was used for detection of Δpro[31]. Secondary antibodies were purchased from Santa Cruz, except that against immunoprecipitated samples, which was from Abcam.

**Western blot analysis**. To prepare the nuclear fraction, tissues after quick excision were homogenized in 2 M sucrose buffer, containing 10 mM HEPES at pH 7.9, 25 mM KCl, 1 mM EDTA, 10% glycerol, 0.15 mM spermine, 2 mM spermidine, and COMPLETE (Roche), followed by ultracentrifugation using Optima LE-80K Ultracentrifuge (Beckman Coulter) and the SW28 rotor (100,000 × g for 1 h)[55]. To prepare the microsomal fraction, fresh tissues were homogenized in 0.25 M sucrose buffer, containing 10 mM Tris-HCl at pH 7.4 and 1 mM EDTA, followed by centrifugation at 900 × g for 10 min, and the supernatant was subjected to ultracentrifugation using the SW28 rotor (100,000 × g for 1 h)[56]. The pellets were resuspended in the liver buffer (25 mM Tris-HCl at pH 7.4, 10 mM sodium orthovanadate, 10 mM sodium pyrophosphate, 100 mM sodium fluoride, 10 mM EDTA, 10 mM EGTA, and COMPLETE). To prepare total lysates, frozen tissues were homogenized in the liver buffer and subjected to ultracentrifugation using the 70.1Ti rotor (280,000 × g for 1 h)[6]. The samples were resolved on SDS-PAGE and transferred to Hybond-P membranes (GE Healthcare)[6]. The membranes were incubated with primary antibodies, after blocking with bovine serum albumin (Sigma) resolved in Tris Buffered Saline (TBS) with Tween 20 (anti-pY, anti-p-S6K, anti-p-4E-BP-1, and anti-p-mTOR antibodies), blocking solution for DIG (Roche) diluted in TBS with Tween 20 (anti-Ub, anti-calnexin, anti-p-PERK, anti-p-Foxo1, and anti-TMED10 antibodies), blocking solution for DIG diluted in TBS with Triton X-100 (WAKO) (anti-XBP-1 and anti-ATF6 antibodies), skim milk (Megmilk Snow Brand) resolved in TBS with Tween 20 (anti-HA and anti-RNAP I antibodies), or skim milk (WAKO) resolved in TBS with Triton X-100 (the other antibodies). Phosphorylation of PERK in the liver was detected carefully with the samples subjected to overnight electrophoresis and the membranes washed carefully, following a previous report[57]. Bound primary antibodies were detected with HRP-conjugated secondary antibodies, using ECL detection reagents (GE Healthcare). Uncropped images are shown in Supplementary Fig. 11.

**Plasmids**. *Sdf2l1* and *Hspa5* cDNA were amplified by PCR with murine liver cDNA as the template, and inserted into pcDNA3.1 (Invitrogen). For restoration of Pmt2-Sdf2l1 fusion protein in yeast, *Sdf2l1* cDNA was inserted into

**Table 2 Primers**

| | | |
|---|---|---|
| Sdf2l1-floxed, external probe | Forward | CCTGGTGTGATGAGCTTGG |
| | Reverse | GGTCTTGGAAGATCTAGAGCC |
| Primers for genotyping | | |
| Sdf2l1-floxed | Forward | CCACCGGTACTTAGGGTCTG |
| | Reverse | CACCCCGTGGATTCTTGTTA |
| Sdf2l1-floxed, ΔNeo | Forward | ACAGAGGGAGGGATGTAGGG |
| | Reverse | TCTTTAAGACCCAGCCCTGA |
| Primers for RT-PCR, murine genes | | |
| Ppia3 | Forward | GGTCCTGGCATCTTGTCCAT |
| | Reverse | CAGTCTTGGCAGTGCAGATAAAA |
| | Probe | CTGGACCAAACACAAACGGTTCCCA |
| SREBP1c[48] | Forward | AAGCTGTCGGGGTAGCGTC |
| | Reverse | TGAGCTGGAGCATGTCTTCAA |
| | Probe | ACCACGGAGCCATGGATTGCACATT |
| Dnajb11 | Forward | CAACTGTCGGCAAGAGATGA |
| | Reverse | CGGCTCACCTTCTCCAATAA |
| sXbp1[49] | Forward | CTGAGTCCGAATCAGGTGCAG |
| | Reverse | GTCCATGGGAAGATGTTCTGG |
| uXbp1[49] | Forward | CAGCACTCAGACTATGTGCA |
| | Reverse | =sXbp1 Reverse |
| Hyou1 | Forward | ACAGATTGAGGGCTTGATGG |
| | Reverse | AACTTTGGGAACACGAGTGG |
| Syvn1 (PrimerBank ID 258547103c1) | Forward | CTCATGCCTACTACCTCAAACAC |
| | Reverse | TGCCCGAAGAACACCTTGC |
| Pomt1[50] | Forward | CTACATCCCAGGACCAGTGCTCAGA |
| | Reverse | AGCGGGACCAGGCATCCTCA |
| Pomt2[50] | Forward | TCCAGCATGTTGACAGGTATCCTATGG |
| | Reverse | CATAAGCCAGAGGGTGGAAGAGGTAGA |
| Tmed10 | Forward | TTGCCTTTACCACGGAAGAC |
| | Reverse | ACTCCACCTCCAGTGGTTTG |
| Mbtps1[51] | Forward | TGCTCCCACCTGACTTTGAAG |
| | Reverse | GCTGTGAAGTATCCGTTGAAAGC |
| Scap[51] | Forward | ATTTGCTCACCGTGGAGATGTT |
| | Reverse | GAAGTCATCCAGGCCACTACTAATG |
| Sdf2l1 ROI | Forward | CACACGGTCCAATAGCAGTG |
| | Reverse | GCTCTAGACCTCTGCGCTTC |
| Sdf2l1 -3.5 kb | Forward | CTGGCCTTTGACCTCTCTTC |
| | Reverse | ACTTGGCAATGGGAACTGTC |
| Hspa5 ERSE | Forward | GGCCGTTAAGAATGACCAGT |
| | Reverse | TCCAGGTCAGTGTTGTCTCG |
| Hspa5 -4.5 kb | Forward | CAGTATTTCCTGGGCCTTCA |
| | Reverse | TTAGGAACTGGGCTGGAGAA |
| Primers for RT-PCR, human genes | | |
| HSPA5[52] | Forward | CGTGGAATGACCCGTCTGTG |
| | Reverse | CTGCCGTAGGCTCGTTGATG |
| SDF2L1[53] | Forward | CTTACGGGCAAGAACCTG |
| | Reverse | GCACTGTCCATAGGTCCA |
| sXBP1 | Forward | =murine sXbp1 Forward |
| | Reverse | ATCCATGGGGAGATGTTCTGG |
| GAPDH | Forward | GGCTGCTTTTAACTCTGGTA |
| | Reverse | GACTGTGGTCATGAGTCCTT |

pGPD416-Pmt2-HA, using a GeneArt Seamless Cloning and Assembly Kit (Life Technologies) (Supplementary Fig. 3i). The 211 amino acids of Pmt2, from R340 through to K550, included in the loop 5, which is the active center of O-mannosyltransferase[58], was substituted for the 188 amino acids of Sdf2l1, from S29 through to T216. We used BAC Subcloning Kit (Gene Bridges) for cloning of the promoter region of Sdf2l1 from MSM Mouse BAC clone MSMg01-540H18, which was provided by the DNA Bank, RIKEN BioResource Center with the support of The National Bio-Resources Project of the Ministry of Education, Culture, Sports, Science and Technology of Japan (MEXT)[59]. It was inserted into pGL3-Basic (Promega), followed by generation of deletion mutants, using KOD-Plus-Mutagenesis Kit (TOYOBO). Cells were transfected with plasmids using TranIT-LT (Mirus).

**Recombinant protein**. Sdf2l1 cDNA, lacking signal peptide and HDEL sequence, was inserted into pQE-60 (Qiagen). His-tagged recombinant Sdf2l1 expressed in

BL21(DE) (Novagen) was purified with a ProBand Purification System (Invitrogen), after denaturation with imidazole.

**RNA interference**. siRNAs were purchased from Ambion (Silencer Pre-designed siRNA; rat Xbp1: s144588, rat Atf6: s156439, mouse Sdf2l1: 182267, mouse Tmed10: s123749, mouse Syvn1: s92311), except siRNA targeting mouse Hspa5, which was from Invitrogen (Stealth RNAi, Hspa5MSS204938), and transfected in cultured cells using Lipofectamine RNAiMax Reagent (Life Technologies), according to the manufacturer's instructions. Dr. Makoto Miyagishi (National Center for Global Health and Medicine) kindly designed the shRNA for Sdf2l1 whose sequence was GGGTTACGGTGAACTTTGA (mutation underlined), with the loop sequence of GTGTGCTGTCC[60], which was inserted into pENTR4-PUR-hU6beta-icas-normal, a kind gift from Dr. Sahohime Matsumoto (The University of Tokyo).

**Adenoviruses**. All the adenoviruses were generated using AdEasy Adenoviral Vector System (Agilent Technologies), except Ad-CAG-Sdf2l1, which was

generated using Adenovirus Dual Expression Kit (TaKaRa). In experiments in vitro, cultured cells were incubated with crude adenoviruses diluted with Opti-MEM I (GIBCO). In experiments in vivo, adenoviruses were purified by CsCl gradient centrifugation using the SW28 rotor (113,000 ×$g$ for 2 h) and the SW41 rotor (210,000 ×$g$ for 3 h), followed by dialysis against 10% glycerol in PBS. The purified adenovirus was administered intravenously, and the mice were analyzed 7–10 days after the administration, unless otherwise indicated.

**Yeast culture**. Yeast cells of the KNY51 (*der1Δ, pmt2Δ, GAL1-Δpro*) strain were grown in YP-rich medium (YPD + 5xAde: 1% yeast extract, 1% polypeptone, 100 mg/L of adenine hydrochloride, and 2% glucose or 2% galactose) or synthetic complete medium (0.67% yeast nitrogen base without amino acids, 100 mg/L of adenine hydrochloride, all standard amino acids [drop out mix], and 2% glucose or 2% galactose). After transformation with Pmt2-expressing plasmids and induction of Δpro expression by galactose overnight, total cell lysates were extracted for Western blotting[31].

**Promoter assay and in vivo imaging**. Promoter assays in vitro were performed using a Dual-Luciferase Reporter Assay System (Promega) and Lumat LB 9507 (EG&G Berthold), according to the manufacturers' instructions. In vivo imaging was performed using IVIS Lumina LT (Perkin Elmer), after intraperitoneal administration of D-Luciferin Sodium (WAKO) at the dose of 150 mg/kg BW, and promoter activity was quantified by total counts divided by exposure time.

**ChIP assay**. Tissue homogenates were crosslinked with formaldehyde, sonicated for fragmentation of chromatin, and incubated with Dynabeads Protein A (Dynal Biotech), as well as antibodies. DNA was eluted, purified using the MinElute PCR purification kit (Qiagen), and subjected to RT-PCR[61].

**Histological analysis of murine tissues**. Murine tissues were fixed with 4% paraformaldehyde, immersed in sucrose solution for cryoprotection, and embedded in optimal cutting temperature (OCT) compound (Sakura Finetek Japan). The frozen sections (6 mm) were stained by the standard Oil Red O staining procedure.

**Hepatic triglyceride content**. Hepatic triglyceride was extracted from the liver homogenate with Folch solution (chloroform and methanol in a 2:1 ratio), resuspended in ethanol containing Triton X-100, and measured using a quantification kit (WAKO)[62].

**Mass spectrometry**. Samples were processed robotically using a RelyOn ProGest instrument. Gel bands were reduced with dithiothreitol, alkylated with iodoacetamide and digested with trypsin. Peptide solutions were subjected for 1-h LC/MS/MS with a Waters NanoAcquity HPLC system interfaced to a ThermoFisher Q Exactive.

**Human subjects**. From November 2011 to March 2014, at the University of Tokyo Hospital, we prospectively recruited patients with clinically suspected NAFLD, and 301 patients consented to liver biopsy, all of whom gave informed written consent. On the day of admission, comorbid illness and drug intake were recorded, and body height, as well as body weight were measured. Percutaneous liver biopsy was performed using a 16 G needle with a biopsy specimen notch of 20 mm, 5 h after a 75 g oral glucose tolerance test, following an overnight fasting. Blood samples before the tolerance test were subjected to laboratory tests, including AST, ALT, and hemoglobin A1c (HbA1c), as well as fasting blood glucose and fasting immunoreactive insulin[63]. Liver histologic specimens were examined by pathologists blinded to the clinical data and the design of this study, and assessed according to Matteoni's classification[64]. After excluding patients in whom NAFLD was not diagnosed, specimens were scored in the remaining patients, according to the Nonalcoholic Steatohepatitis Clinical Research Network criteria[65]. In subjects who agreed to gene expression analysis with informed written consent among the 301 patients, 3 mm of the liver biopsy samples per subject were spared to measure gene expression if the total length of liver biopsy samples was longer than 18 mm. In this study on ER stress responses, we excluded women and those with poor glycemic control (HbA1c over 8%). This study was approved by the University of Tokyo Medical Research Center Ethics Committee. All relevant ethical guidelines were followed.

**Statistics**. Values of the data are expressed as mean ± SEM, and statistical significance is displayed as $P < 0.05$ (one asterisk), $P < 0.01$ (two asterisks), or $P < 0.001$ (three asterisks, if necessary) in figures. In the analysis of mice and cultured cells, differences between 2 groups were assessed using unpaired 2-tailed $t$-test, unless otherwise indicated. Those among 3 or more groups were assessed using one-way ANOVA (ANalysis Of VAriance) with post-hoc Tukey's Honest Significant Difference (HSD) in EZ-R[66]. In the analysis of human subjects, differences between 2 groups were assessed using Mann–Whitney's $U$ test. Correlation of metabolic parameters or liver histology score with gene expression was examined using Spearman's rank correlation method in EZ-R. Matched control pairs were selected using the optmatch package in EZ-R.

## Data availability
The authors declare that all data supporting the findings of this study are available within the manuscript and its Supplementary Information files or are available from the authors upon reasonable request. A reporting summary for this study is available as a supplementary information file.

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

## Acknowledgements

We thank Norie Ohtsuka-Kowatari, Kayo Maruyama-Nishitani, Ritsuko Hoshino, Kuniko Kurihara, Fumiya Takahashi, Yuko Kanto, Yasuko Sakuma, Reiko Homma, Yuko Masaki, Asako Yoshida, Yoshiko Ito, Mizuki Chosa, Chika Goto, Yasuko Ota, Tomoyuki Ogiwara, Hiroshi Chiyonobu, Katsuyoshi Kumagai, Naoki Ishikawa, Kayo Aoyama, Tokiko Hirabayashi, Ayumi Ohuchi, Yuka Kobayashi, Rie Onoue, and Yumiko Kishida for their excellent technical assistance and assistance with the animal care. We thank Prof. Toshiya Endo (Kyoto Sangyo University) and Prof. Shuh-ichi Nishikawa (Niigata University) for giving us advice on O-mannosylation of proteins in yeast; Dr. Kyohei Umebayashi (The University of Geneva), Dr. Ryoichi Fukuda (The University of Tokyo), and Dr. Toshiaki Izawa (Kyushu University) for providing us the assay system of O-mannosylated Δpro; Prof. Tamao Endo and Dr. Hiroshi Manya (Tokyo Metropolitan Institution of Gerontology) for giving us advice on O-mannosylation of proteins in mammals and microsomal fractionation of tissues; and Drs. Naoya Yahagi and Yoshinori Takeuchi (The University of Tsukuba) for giving us advice on nuclear fractionation of tissues. This work was supported by a grant for Translational Systems Biology and Medicine Initiative (TSBMI), Creation of Innovation Centers for Advanced Inter-disciplinary Research Areas Program of the MEXT; Health and Labour Sciences Research Grants (H20-Kanen-Ippan-008, H24-Kanen-Ippan-006, both to K. U.) by the Ministry of Health, Labour and Welfare of Japan; a grant for the Research Program on Hepatitis from Japan Agency for Medical Research and Development (JP17fk0210304, JP18fk0210040, both to K. U.); a Grant-in-Aid for Challenging Exploratory Research (21659227, to K. U.), Grants-in-aid for Young Scientists (B) (22790851, 24790913, both to T. S.), and a Grant-in-Aid for Scientific Research (C) (26461358, to T. S.) by the MEXT; grants for Front Runner of Future Diabetes Research by the Japan Foundation for Applied Enzymology (to T. S.); and grants and endowments from Daiichi Sankyo Co, Ltd.

## Author contributions

T.S. developed the hypothesis, designed and performed the experiments, analyzed the data, and wrote the manuscript. S.I., A.T., T. Kubota., S.Y., K.N., K.I., H.K., and K.O. performed the experiments. M.O., N.K., Y.O., T. Kamura, K. Tokuyama., Y.F., M.F., T.T., R.T., H.A., and K.K. designed the experiments. J.S. scored histology of human liver specimens. K.E. recruited patients, collected clinical data and performed liver biopsies. K. Tobe developed the hypothesis. T. Kadowaki and K.U. developed the hypothesis, designed the experiments, analyzed the data, and wrote the manuscript.

## Additional information

**Competing interests:** The authors declare no competing interests.

