## [Peer Review File · Nature Communications]

Reviewers' comments:

Reviewer #1 (Remarks to the Author):

In this study, the authors identified an ER-resident molecule, stromal cell-derived factor 2 like 1 (Sdf2l1), as a physiological regulator of ER stress responses and energy (lipid and glucose) homeostasis in the liver. They demonstrated that suppression of Sdf2l1 sustained ER stress, leading to insulin resistance and hepatic steatosis. These are interesting observations that suggest a novel link between ER stress and both insulin resistance and fatty liver disease. The major concerns on this work include:

Regarding "Feeding induces ER stress responses in the liver", in Figure 1, there lacks a control in regard to the basal level of expression or activation of ER stress markers or transducers in the liver of mice under normal chow. Without this critical control, it may be misleading to conclude refeeding causes ER stress response by simply comparing to the induction of ER stress markers or mediators under the fasting condition. It could be the case that refeeding may just restore the induction of ER stress markers or mediators that were suppressed by the fasting condition. This concern applies to the qPCR, Western blot analyses, and ChIP assay as presented in Figure 1 and beyond.

Many supplemental figures are essential evidence of the work. They should be incorporated into the formal figures.

Oil-red O staining (Figure 3K) should be accompanied with quantitative measurement of hepatic triglycerides by an enzymatic assay.

The figures were not carefully formalized or organized. For example, in Figure 5, the statistical "*" were not carefully put in place with the figure graphs.

The information about the diabetic patient background and resources was not sufficient.

Reviewer #2 (Remarks to the Author):

Sasako et al surveyed the gene expression profile in mouse liver upon refeeding and confirmed a transient elevation in the expression of ER stress responsive genes, one of which is Sdf2l1. In vitro analysis showed that Sdf2l1 expression is regulated by IRE1-Xbp-1 and Sdf2l1 interacts with TMED10, a component in ERAD complex. They further observed that transient suppression of Sdf2l1 using adenovirus-mediated shRNA knockdown resulted in a partial insulin resistance and increased triglyceride, implying a possible involvement of Sdf2l1 downregulation in metabolic disorders. Indeed, Sdf2l1 downregulation is detected in liver tissues from mice and human with obesity or diabetes. Those observations imply Sdf2l1 as a potentially important factor in liver metabolic regulation and therapeutic target to treat metabolic disorders.

1. However, the main conclusion solely relies on a transient (7-10 days) suppression of Sdf2l1 using adenovirus-mediated shRNA knockdown, which only last for few days. While the manuscript claimed that a strain of Sdf2l1 floxed mice has been successfully established, it is surprising to this reviewer that this strain of mice were not used in the study. Therefore, the main conclusion should be further validated using mice with liver-specific targeted Sdf2l1 deletion (either permanent or inducible), in particular after a long-term HFD feeding, to warrant its publication by Nature Communications.

2. In line with #1, it is unclear whether a long-term or permanent Sdf2l1 suppression/deletion is toxic to liver in mice.

3. The authors used adenovirus mediated knockdown (Fig. 3) or expression (Fig. 5) of Sdf2l1 for the studies. This is preferably done using AAV-mediated knockdown system because adenovirus triggers more profound inflammation and other side-effects. And more importantly, adenovirus mediated knockdown or expression is transient and only lasts for days, in contrast AAV- mediated knockdown or expression often lasts for months. This is in particular important for studying the therapeutic effects of restoring hepatic expression of Sdf2l1, which allows to observe Sdf2l1 therapy in a optimal period to treat chronic diseases such as obesity and type II diabetes.
4. The authors detected a substantial increase in p-IRE1 α , the only known enzyme that splices Xbp-1 mRNA, in db/db mouse liver (Fig. 4a), but only a modest increase in the levels of Xbp-1s was detected 6 hours after refeeding but not at any earlier time point. Whether Xbp-1 splicing occurs at the later time point? If not, what is the explanation?
5. The authors claims that Xbp-1s nuclear translocation is impaired in db/db mouse liver, but the data in Fig, 4c is not convincing, which only show a modest reduction at 1 hour after refeeding but not any other time points. For example, 3 hours after refeeding, Xbp-1s nuclear translocation is indistinguishable. However, a dramatic reduction in Xbp-1s binding to Sdf2l1 promoter was detected 24 hours after refeeding. Is this because nuclear Xbp-1s is further reduce during the later time points after refeeding? The authors should provide data to show the nuclear Xbp-1s levels in line with its promoter binding activity at the same time point.
6. Also, Fig. 4c, the total and cytoplasmic Xbp-1s and nATF6 should be shown as controls to support the conclusion that Xbp-1s and nATF6 nuclear translocation (but not their levels) is impaired in db/db mouse liver.
7. While the authors claimed that the liver lipid deposition was dramatically increased by Sdf2l1 knockdown, data in Fig. 3k are not convincing, this is possibly due to a transient knockdown within few days. As described in concern #1, fatty liver disease should be further characyerized in a genetic model after a longer term of HFD.
8. The authors defined Sdf2l1-TMED10 ERAD complex that catalyzes ubiquitination of misfolded protein for degradation (in this case insulin, Fig. 2), however they never looked at other factors involved in this ERAD complex in particular which E3 ligase is responsible for this process. It could be that Sdf2l1-TMED10 complex acts independent of the canonical ERAD pathway. This point is very important and needs further clarification. Other potential effects caused by Sdf2l1 suppression, such as on ER-Golgi protein transportation and processing should be considered.
9. While the authors described that 301 liver biopsy samples were collected in the methods section, only 64 male subjects were studied. A justification to exclusively use male samples should be provided. It is also important to show whether a same conclusion can be obtained if female samples are used.

Outline of Changes and Responses to the Comments Made by the Reviewers

Responses to the comments made by the Reviewer 1:

We are extremely grateful to the Reviewer 1 for the very careful review of our manuscript and for the suggestions that made our manuscript markedly improved.

> Regarding “Feeding induces ER stress responses in the liver”, in Figure 1, there lacks a control in regard to the basal level of expression or activation of ER stress markers or transducers in the liver of mice under normal chow. Without this critical control, it may be misleading to conclude refeeding causes ER stress response by simply comparing to the induction of ER stress markers or mediators under the fasting condition. It could be the case that refeeding may just restore the induction of ER stress markers or mediators that were suppressed by the fasting condition. This concern applies to the qPCR, Western blot analyses, and ChIP assay as presented in Figure 1 and beyond.

We first showed the expression of ER stress marker genes in an *ad libitum*-fed state, as the time point of ‘-24h’, in Figure 1a in the original manuscript. They were highly up-regulated during refeeding compared to during fasting, and thus we focused on the refeed state thereafter. Besides, we examined phosphorylation of IRE1a, in accordance with the suggestion made by the reviewer, which was a little lowered by fasting and highly elevated by refeeding (Supplementary Fig. 1b).

We modified the manuscript, so that the readers could easily get the point, as follows:

Results

Although expression of ER stress marker genes, including classical ones, namely spliced Xbp1 (*sXbp1*, encoding XBP-1s), *Hspa5*, and *Ddit3*, as well as *Sdf211*, showed little change during fasting compared to the *ad libitum* condition, it was elevated prominently and transiently after refeeding (Fig. 1a and Supplementary Fig. 1a). Thus we focused on the refeed state thereafter, and at the protein level, phosphorylation, expression, and nuclear localization of ER stress markers, as well as *Sdf211*, were also elevated during refeeding, as analyzed by total lysates (Fig. 1b and Supplementary Fig. 1b, c), immunoprecipitates (Supplementary Fig. 1d), and nuclear extracts (Fig. 1c). (pp.6-7, ll. 126-134)

Figure Legends

Figure 1

(a-d) C57BL/6J mice in an *ad libitum*-fed state were fasted for 24 hours and then refed for 6 hours (n = 3-4), and ER stress markers were analyzed by (a) RT-PCR, (b) Western blotting of total lysates, (c) Western blotting of nuclear extracts (XBP-1s and nATF6 were detected at 55 and 50 kDa, respectively), and (d) Western blotting of microsomal fractions. (p.41 , ll. 904-908)

As described in the manuscript, disorders of the metabolic regulations during the transition between fasting and feeding in the liver may be the key players of obesity-related diseases, and many studies have compared the changes between fasting and refeeding in the liver. Indeed our protocol followed those in the previous reports that focused on the refed state after fasting for 24 hours (Ref. 6, 8, 19). Moreover, ER stress response was susceptible to food intake during refeeding, which much varied in a relatively short fasting, and thus 24-hour fasting was technically desirable.

> Many supplemental figures are essential evidence of the work. They should be incorporated into the formal figures.

In accordance with the reviewer's comment, we further analyzed the knockout model of *Sdf211* in an independent formal figure (Figure 4).

> Oil-red O staining (Figure 3K) should be accompanied with quantitative measurement of hepatic triglycerides by an enzymatic assay.

We already showed quantified triglyceride contents extracted from liver samples in the original manuscript (Figure 3j), and confirmed the consistency with the results of Oil Red O staining. Accordingly, we also quantified triglyceride contents in the knockout model of *Sdf211* (Figure 4h), in addition to the results of Oil Red O staining (Figure 4i) which had been included in the original manuscript.

> The figures were not carefully formalized or organized. For example, in Figure 5, the statistical “*” were not carefully put in place with the figure graphs.

We had shown the asterisks at a fixed height in bar graphs, but rearranged them in all kinds of graphs in accordance with the suggestion made by the reviewer.

> The information about the diabetic patient background and resources was not sufficient.

In accordance with the suggestion of the reviewer, we analyzed the background characteristics separately in subjects with diabetes and those without. It was natural that those with diabetes showed higher HbA1c and HOMA-R, but they were older and showed advanced stage of NASH (Table 1). Therefore, among the 39 nondiabetic subjects, we selected 25 as those matched with the 25 diabetic subjects based on age and histological findings, i.e. NAS and stage (Table 1). Instead of the results of analyses in the 39 pre-match nondiabetic subjects in the original manuscript, we included those in the 25 post-match nondiabetic subjects (Figure 7c-h), recapturing the results using the whole subjects. These data further confirmed the pathophysiological roles of ER stress ‘response failure’ in a manner specific to diabetes-associated NASH.

We also added description in the Results and Methods, as follows:

Results

It was first revealed that expression of SDF2L1 was negatively correlated with glycemic control (Fig. 7b), and we focused on the presence or absence of diabetes thereafter. Although it was natural that the diabetic subjects showed higher HbA1c and HOMA-R, they were older and showed advanced stage of NASH. Therefore, among the 39 nondiabetic subjects, we selected 25 as those matched with the 25 diabetic subjects based on age and histological findings, i.e. NAS and stage (Table 1). Expression of SDF2L1 was significantly down-regulated in those with diabetes (Fig. 7c), whereas that of the other genes was not (data not shown). (p. 14, ll. 316-323)

Methods

Statistics

Matched control pairs were selected using the optmatch package in EZ-R. (p. 30, ll. 682-683)

Responses to the comments made by the Reviewer 2:

We are extremely grateful to the Reviewer 2 for the very careful review of our manuscript and for the suggestions that made our manuscript markedly improved.

> 1. However, the main conclusion solely relies on a transient (7-10 days) suppression of *Sdf211* using adenovirus-mediated shRNA knockdown, which only last for few days. While the manuscript claimed that a strain of *Sdf211* floxed mice has been successfully established, it is surprising to this reviewer that this strain of mice were not used in the study. Therefore, the main conclusion should be further validated using mice with liver-specific targeted *Sdf211* deletion (either permanent or inducible), in particular after a long-term HFD feeding, to warrant its publication by Nature Communications.

> 7. While the authors claimed that the liver lipid deposition was dramatically increased by *Sdf211* knockdown, data in Fig. 3k are not convincing, this is possibly due to a transient knockdown within few days. As described in concern #1, fatty liver disease should be further characterized in a genetic model after a longer term of HFD.

We appreciate the comment made by the reviewer, because it is very important to investigate the phenotypes of the liver-specific *Sdf211* knockout mice, and confirm that they are essentially the same as those observed in the knockdown model of *Sdf211*. Actually, we analyzed the phenotypes of the knockdown model intensively because it had been established much earlier than the knockout model.

We also generated the inducible liver-specific knockout mice by administering an adenovirus, Ad-Cre, to *Sdf211*-floxed mice, and already showed some of the phenotypes in the supplementary figure in the original manuscript, such as enhanced ER stress during refeeding (shown in the revised Figure 4c), elevated *ad libitum*-fed plasma glucose (shown in the revised Figure 4d), glucose intolerance (shown in the revised Figure 4g), and increased lipid accumulation (shown in the revised Figure 4i). Following the reviewer's suggestion, We now show these data in an independent formal figure (Figure 4). Besides, we further analyzed the phenotypes, including systemic insulin resistance (Figure 4e), elevated gluconeogenesis (Figure 4f), up-regulation in another ER stress marker gene (Figure 4c), and increased triglyceride contents (Figure 4h) in the model, despite unaltered body weight (Figure 4b), and these phenotypes replicated those observed by adenovirus-mediated transient knocking down of *Sdf211* (Figure 3 and Supplementary Figure 6). It should

be noted that the same titer of the control virus was administered to the control mice, and importantly, these analyses were performed 3-6 weeks after the administration, which was probably long enough to eliminate, if any, toxic effects of adenovirus administration.

We clarified these points, including the time course of the experiments, as follows:

Results

We also generated liver-specific *Sdf211*-knockout mice by administering adenovirus expressing Cre recombinase to *Sdf211*-floxed mice (Fig. 4a and Supplementary Fig. 4a-c), which exhibited similar phenotypes for a longer period (Fig. 4b-i), further supporting data of the knocking down model. (p. 11, ll. 251-254)

Methods

Adenoviruses

The purified adenovirus was administered intravenously, and the mice were analyzed 7-10 days after the administration, unless otherwise indicated. (p. 26, ll. 615-617)

Figure Legends

Figure 4

(a) Detection of deleted exon 2 of the *Sdf211* gene by PCR.

(b-i) Metabolic phenotypes analyzed 3-6 weeks after the adenovirus administration (n = 8-9): (b) body weight, (c) ER stress markers analyzed by RT-PCR, (d) ad libitum-fed plasma glucose, (e) plasma glucose in ITT, after intraperitoneal injection of human regular insulin (0.75 U/kg BW), (f) plasma glucose levels in an pyruvate tolerance test, after intraperitoneal injection of pyruvate (1.5 g/kg BW), (g) plasma glucose levels in an OGTT, after oral administration of glucose (0.75 g/kg BW), following 16 hours of fasting, (h) triglyceride contents quantification, and (i) Oil Red O staining. (pp. 43-44, ll. 967-975)

Next, as for the effects of suppressed expression of *Sdf211* in metabolism in mice fed with high-fat diet, we also fed the adenovirus-mediated liver-specific *Sdf211* knockout mice with high-fat diet, as was suggested by the reviewer. Ad-Cre was administered to *Sdf211*-floxed mice, with high-fat diet feeding started

simultaneously, and these mice also exhibited glucose intolerance, even 7 weeks after the administration (Figure 1 Only for the Reviewers), just as the knockdown and knockout models of *Sdf211* fed with normal chow (Fig. 3f and 4g).

We consider it too long to feed liver-specific *Sdf211* knockout mice with high-fat diet for more than 2 months. As was shown in Supplementary Figure 7e, expression of *Sdf211* in a refed state was not down-regulated in mice fed with high-fat diet for 8 weeks, a model of insulin resistance without overt hyperglycemia. However, high-fat diet feeding for a longer period is expected to suppress expression of *Sdf211*, as insulin resistance becomes decompensatory, just as seen in *db/db* mice, a model of severe insulin resistance and hyperglycemia (Figure 5). Indeed, *Sdf211* expression was suppressed after 12 weeks of high-fat diet feeding (Figure 2 Only for the Reviewers), which would diminish the effects of knocking out of *Sdf211*. Therefore, we consider mice with *Sdf211* knocked down or knocked out on normal chow, rather than on high-fat diet, as models to mimic ER stress ‘response failure’ observed in severe obesity.

We described the details of the diet-induced obesity model in the supplementary figure, as follows:

Supplementary Figure Legends

Supplementary Fig. 7

(d, e) ER stress markers analyzed by RT-PCR in (d) *ob/ob* mice, and (e) mice fed with high-fat diet for 8 weeks (n = 4 for each).

Overall, the phenotypes observed in the knockdown model for a relatively short period were replicated by those observed in the knockout model for a longer period, and we believe that these data altogether prove the roles of *Sdf211* in the regulation of metabolism in the liver. Moreover, it is very important that these results of experiments in mice were replicated by those in human liver samples, proving that chronic ER stress ‘response failure’ links insulin resistance and the development of NASH associated with diabetes.

> 2. In line with #1, it is unclear whether a long-term or permanent *Sdf211* suppression/deletion is toxic to liver in mice.

In mice with *Sdf211* knocked down or knocked out, no histological abnormal

findings were observed in the liver, except fatty changes (Fig. 3k and 4i). Still it is probable that morphological abnormalities could be found in the ER, as was pointed out by the reviewer, which remains to be clarified in future works.

> 3. The authors used adenovirus mediated knockdown (Fig. 3) or expression (Fig. 5) of *Sdf211* for the studies. This is preferably done using AAV-mediated knockdown system because adenovirus triggers more profound inflammation and other side-effects. And more importantly, adenovirus mediated knockdown or expression is transient and only lasts for days, in contrast AAV-mediated knockdown or expression often lasts for months. This is in particular important for studying the therapeutic effects of restoring hepatic expression of *Sdf211*, which allows to observe *Sdf211* therapy in a optimal period to treat chronic diseases such as obesity and type II diabetes.

As was argued in our response to the first point made by the Reviewer 2, we generated *Sdf211* knockout mice, instead of an AAV-mediated knockdown model, and confirmed that their phenotypes were similar to those observed in the adenovirus-mediated knockdown model for a longer period. We also discussed potential toxic effects of administration of adenoviruses.

As for restoration of *Sdf211* in *db/db* mice, it is an important finding and should be noted that severe metabolic disorders of *db/db* mice were improved by just transient restoration of *Sdf211* only for 7-10 days. Besides, we believe that it is very intriguing that restoration of the downstream molecules including *Sdf211* is beneficial, but restoration of the upstream transcription factor itself is not enough, which would contribute to focusing on potential targets of treatment to modulate ER stress response in the liver. We totally agree with the reviewer that it is an important issue whether such beneficial effects could be sustained for a longer period, which could be addressed in future works using AAV-mediated restoration or inducible transgenic mice, and discussed these points, as follows:

Discussion

Based on our data, this may be partly through efficient induction of *Sdf211* and BiP by XBP-1s, while over-expression of XBP-1s alone is not enough, because the translocation to the nucleus remains impaired without improving insulin resistance. It remains to be clarified whether the beneficial effects of

Sdf211 restoration could be sustained for a longer period, for weeks or months. Besides, to develop more effective drugs for these diseases by shutting down the vicious cycle, it may be useful to identify the effectors downstream of Sdf211 to regulate ERAD, besides the strategy to promote the translocation of XBP-1s to the nucleus to up-regulate Sdf211 ^{20,38}. (p. 17, ll. 395-403)

> 4. The authors detected a substantial increase in p-IRE1 α , the only known enzyme that splices Xbp-1 mRNA, in db/db mouse liver (Fig. 4a), but only a modest increase in the levels of Xbp-1s was detected 6 hours after refeeding but not at any earlier time point. Whether Xbp-1 splicing occurs at the later time point? If not, what is the explanation?

As was pointed out by the reviewer, splicing of *sXbp1* mRNA by IRE1 α was increased 6 hours after refeeding but not at earlier time points. This could be also due to a discrepancy between function and phosphorylation of IRE1 α by some reasons associated with obesity or diabetes. Similarly, nuclear translocation of XBP-1s could be also delayed (Figure 5c), possibly contributing to the development of ER stress ‘response failure’ in the liver. As we discussed in the Discussion, it might be unique to the liver in obesity and diabetes, which should be investigated in future works.

We added description on this point, as follows:

Results

As for the IRE1 α -ATF6 branch, *sXbp1* mRNA was up-regulated only slightly (Fig. 5b), possibly suggesting impaired and delayed splicing activity of IRE1 α . XBP-1s and nATF6 located in the nucleus, as well as total XBP-1s and nATF6, were significantly decreased early in the refed state (Fig. 5c), as we and others have previously reported ^{19,20,38}. (p. 12, ll. 261-265)

Among them, Sdf211 was most prominently down-regulated both during fasting and feeding, accompanied by delayed nuclear localization of XBP-1s during refeeding, presumably due to the decreased insulin action to promote the translocation of XBP-1s by binding to p85 ^{20,38}. (p. 12, ll. 276-280)

Discussion

Insulin resistance in the first hit can cause dysfunction of XBP-1s, presumably due to delayed insulin-mediated translocation to the nucleus^{20,38}, resulting in the suppression of the termination signal for ER stress such as induction of SDF2L1, leading to sustained ER stress that can function as a second hit. (p. 16, ll. 376-380)

> 5. The authors claims that Xbp-1s nuclear translocation is impaired in db/db mouse liver, but the data in Fig, 4c is not convincing, which only show a modest reduction at 1 hour after refeeding but not any other time points. For example, 3 hours after refeeding, Xbp-1s nuclear translocation is indistinguishable. However, a dramatic reduction in Xbp-1s binding to Sdf2l1 promoter was detected 24 hours after refeeding. Is this because nuclear Xbp-1s is further reduce during the later time points after refeeding? The authors should provide data to show the nuclear Xbp-1s levels in line with its promoter binding activity at the same time point.

Figure 5d showed the results of ChIP assay “in a 1-hour refed state after” 24 hours of fasting, as was stated in the figure legend in the original manuscript, the time point when XBP-1s in the nucleus was decreased.

We added the time point in the graphs, so that the readers could easily get the point (Figure 5d and Supplementary Figure 7c).

> 6. Also, Fig. 4c, the total and cytoplasmic Xbp-1s and nATF6 should be shown as controls to support the conclusion that Xbp-1s and nATF6 nuclear translocation (but not their levels) is impaired in db/db mouse liver.

It should be noted that the concept of impaired nuclear translocation of XBP-1s and ATF6 in obesity and diabetes was mainly proven by experiments to forcedly express the factors *in vitro* in the previous reports (Ref. 19, 20). The direct comparison was not performed between the endogenous expression levels of them in the nucleus and those in other fractions *in vivo*: it was shown instead that the expression in the nucleus was decreased in model mice, and importantly, total expression of XBP-1s and ATF6 was also decreased. Probably it was due to enhanced degradation of the transcription factors which failed to be translocated into the nucleus and thus were

unable to exert the transcription activity, while these proteins in the nucleus were relatively stable. Actually, in the other report that first proved that insulin signaling promotes translocation of XBP-1s, it was shown that the signaling stabilized XBP-1s protein (Ref. 38).

Therefore we consider it important to show the expression in the nucleus by Western blotting using nuclear extracts, and binding with the promoter region of the downstream chaperones by ChIP assay, as were included in the original manuscript (Figure 5c, d and Supplementary Figure 7c). Still we agree with the reviewer that ‘nuclear translocation’ should be distinguished from ‘expression in the nucleus’, and modified the description, as follows:

Results

XBP-1s and nATF6 located in the nucleus were significantly decreased early in the refed state (Fig. 4c), as we and others have previously reported^{19,20,37}. (p. 12, ll. 263-265)

> 8. The authors defined Sdf211-TMED10 ERAD complex that catalyzes ubiquitination of misfolded protein for degradation (in this case insulin, Fig. 2), however they never looked at other factors involved in this ERAD complex in particular which E3 ligase is responsible for this process. It could be that Sdf211-TMED10 complex acts independent of the canonical ERAD pathway. This point is very important and needs further clarification. Other potential effects caused by Sdf211 suppression, such as on ER-Golgi protein transportation and processing should be considered.

In accordance with the suggestion made by the reviewer, we explored the roles of Sdf211 in ERAD in more details, and focused on Hrd1, because it is known that Pmt1/2p, the orthologs of Sdf211, promotes Hrd1p-mediated ERAD in yeast (Ref. 32). Hrd1 is also known to play a pivotal role in ERAD as a major E3 ligase in mammals (*Nat Struct Mol Biol* 21, 325-335 (2014): Ref. 37).

Knocking down of *Syvn1* (encoding Hrd1) resulted in enhanced ER stress, which was not affected by additional knocking down of *Sdf211* (Supplementary Figure 5a, b, d). It showed a clear contrast with BiP, because knocking down of *Hspa5* (encoding BiP) resulted in enhanced ER stress, which was further enhanced by additional knocking down of *Sdf211* (Figure 2d). Besides, accumulation of ubiquitinated mutant

insulin by knocking down of *Sdf211* was canceled by additional knocking down of *Syvn1* (Supplementary Figure 5c). Thus it was suggested that Sdf211 could modulate ERAD in a manner dependent on Hrd1, and exert its effect after ubiquitination of substrates for ERAD, possibly by shuttling them for the proteasome. Actually, some chaperones are known to function outside the ER as well, including Dnajb2, which is known to be also involved outside the ER in the shuttling for the proteasome (Ref. 37). Moreover, given that Sdf211 functions outside the ER in terms of promotion of ERAD, it seems consistent with our data in which accumulation of ubiquitinated mutant insulin was not observed by thapsigargin-induced global dysfunction of chaperons in the ER (Figure 2c).

In terms of protein transportation from the ER to the Golgi apparatus, we focused on potential interaction of Sdf211 and β COP, a key component of COPII vesicles involved in the transportation, but it was not observed (Figure 2e), although TMED10 is known to be involved in the transportation (*J Cell Biol* 194, 61-75 (2011): Ref. 36). We also investigated whether β COP expression was affected by knocking down of *Sdf211*, in accordance with the suggestion made by the reviewer, only to find that it was not (Figure 3 Only for the Reviewers). However, although we failed to reveal the clear involvement of Sdf211 in the ER-Golgi protein transportation as a component of COPII vesicles, it is still probable that Sdf211 could traffic between the ER and the Golgi apparatus.

In order to address the question why Sdf211 needs interaction with TMED10, we focused on anchoring in the membrane of the ER. In yeast, Pmt1/2p, the orthologs of Sdf211, are anchored in the ER membrane and interact with p24, the ortholog of TMED10. In mammals, however, Sdf211 lacks a transmembrane domain (Supplementary Figure 3i), and it is probable that Sdf211 requires TMED10 as a counterpart serving as a platform in order to work near or even through the ER membrane.

We included additional data, references, and discussion in the revised manuscript, as follows:

Results

Interestingly, such accumulation was not observed by knocking down of BiP or thapsigargin-induced global dysfunction of the ER (Fig. 2c), although Sdf211 has been reported to bind to BiP in other tissues²⁶⁻²⁸. Moreover, ER stress was further enhanced by additional knocking down of *Hspa5* compared to knocking down of *Sdf211* alone (Fig. 2d). (p. 9, ll. 196-200)

TMED10 is the ortholog of p24, a membrane protein which interacts with Pmt1/2p and promotes ER export of unfolded proteins for ERAD in yeast ³², and is known to be involved in COPII vesicle-mediated protein transportation from the ER to the Golgi apparatus ³⁶. Although TMED10 showed almost constitutive expression patterns, contrary to Sdf211 (Fig. 1d and Supplementary Fig. 1a and 3a), Sdf211 bound to the 24 kDa isoform of TMED10, but not to β COP, a key component of COPII vesicles, in the liver (Fig. 2e). (p. 10, ll. 208-214)

Moreover, it is known that Pmt1/2p promotes Hrd1p-mediated ERAD in yeast ³², and actually also in mice, the effects of knocking down of Sdf211 were mainly dependent on Hrd1 (Supplementary Fig. 5), the major E3 ligase in the canonical pathway of ERAD in mammals ³⁷. (p. 10, ll. 223-226)

In accordance with the findings *in vitro* (Fig. 2d, g), compared to single restoration of Sdf211, additional restoration of BiP, had the larger beneficial effects (Fig. 6i, j and Supplementary Fig. 8f-j), suggesting that Sdf211 could improve insulin sensitivity independently of BiP, at least in part. (p. 13, ll. 301-304)

Discussion

Given that Sdf211 has an ER-retention-like motif, it is expected to traffic between the ER and the Golgi apparatus and cope with ER stress, even during feeding, although Sdf211 is not likely to be a component of COPII vesicles. (p. 15, ll. 340-342)

Moreover, it is suggested that Sdf211 could be involved, outside the ER, in shuttling of substrates for ERAD ubiquitinated mainly by Hrd1 to the proteasome, just as Dnajb2, which works as a chaperone inside the ER and is also involved in the shuttling outside the ER ³⁷. In yeast, Pmt1/2p, the orthologs of Sdf211, are anchored in the ER membrane and interact with p24, the ortholog of TMED10. Sdf211 in mammals, however, lacks a transmembrane domain, and thus it is probable that it requires TMED10 as a counterpart serving as a platform in order to work near or even through the ER membrane. (p. 15, ll. 347-355)

> 9. While the authors described that 301 liver biopsy samples were collected in the methods section, only 64 male subjects were studied. A justification to exclusively use male samples should be provided. It is also important to show whether a same conclusion can be obtained if female samples are used.

Although we understand that it is extremely important to reveal the mechanism to account for the development of NASH also in females, we focused on liver samples of male subjects, in accordance with the experiments in mice which were all performed using male mice. It was also because the pathophysiology of NASH in females is complicated, because it is largely affected by the presence or absence of menopause, but further investigation is surely needed.

We discussed these points in the Discussion, as follows:

Discussion

Therefore, the lower SDF2L1/sXBP1 ratio could be a much better biomarker than other ER stress-related genes, such as sXBP1 alone, to reflect not only ER stress but also ER stress 'response failure' that leads to progression of diabetes-associated diseases in the liver. It should be investigated in future works whether such ER stress 'response failure', observed in male mice and male subjects, could contribute to in the development of NASH in female subjects as well, which is known to be largely affected by menopause. (p. 17, ll. 384-390)

Figure 1 Only for the Reviewers

Ad-cre was administered to *Sdf211*-floxed mice, with high-fat diet feeding started simultaneously, and an OGTT (glucose 0.75 g/kg BW) was performed 7 weeks later.

Figure 2 Only for the Reviewers

ER stress marker gene expression in mice fed with high-fat diet for 12 weeks.

Figure 3 Only for the Reviewers

COPII vesicle marker gene expression in primary hepatocytes with *Sdf211* knocked down.

Reviewers' comments:

Reviewer #1 (Remarks to the Author):

While most comments were addressed, the CHIP assay results, for example in Fig 1H, need to be confirmed/accompanied by gel images showing visible amplification productions through conventional CHIP-PCR. qPCR alone is somewhat not convincing.

Additionally, the labeling and legends in the figures were poorly composed of.

Reviewer #2 (Remarks to the Author):

The authors have fully addressed my concerns. This is an important study, which defines a novel molecular mechanism by which Sdf2l1-ERAD regulates hepatic metabolism and the ER stress 'response failure' in metabolic pathogenesis. Its publication by Nature communication is recommended by this reviewer.

Changes and Responses to the Comments Made by the Reviewer

Responses to the comment made by the Reviewer 1:

We are extremely grateful to the Reviewer 1 for the very careful review of our revised manuscript and also for an additional suggestion.

> While most comments were addressed, the ChIP assay results, for example in Fig 1H, need to be confirmed/accompanied by gel images showing visible amplification productions through conventional ChIP-PCR. qPCR alone is somewhat not convincing.

As was suggested by the reviewer, we performed PCR using the same ChIP samples as had been used for Figure 1h and Supplementary Figure 3h. First, plateau-phase PCR products (50 cycles) were subjected to electrophoresis, and clear single bands were observed, as was expected (Figure 1a Only for the Reviewer). Then log-phase PCR products (XBP-1: 28 cycles, ATF6: 39 cycles) were subjected to electrophoresis, and the binding of XBP-1 and ATF6 with the promoters was increased in the refed state (Figure 1b Only for the Reviewer), further supporting the results of ChIP-qPCR (Figure 1h and Supplementary Figure 3h).

We prefer ChIP-qPCR to qualitative ChIP-PCR, because ChIP-qPCR is more quantitative and less arbitrary. Results of qualitative PCR are affected by cycles of PCR, but those of qPCR is less affected by such PCR conditions. Moreover, in figures to show results of ChIP-qPCR in our manuscript, the relative expression levels were normalized by the upstream region and further by control IgG. For example, the binding of XBP-1 with the region of interest (ROI) of the *Sdf211* promoter was normalized by the upstream region, -3.5kb, and the ROI/-3.5kb ratio was further normalized by that of control rabbit IgG (Figure 1h).

We additionally described such details of ChIP-qPCR, as follows:

Results

Moreover, chromatin immunoprecipitation (ChIP) assays showed that the binding of XBP-1s and nATF6 was elevated during refeeding with the region of interest (ROI) of the *Sdf211* promoter, as well as with the ERSE of the *Hspa5* promoter (Fig. 1h and Supplementary Fig. 3h). These data suggest that ER stress directly induces *Sdf211* expression via XBP-1s and nATF6. (p.8, ll. 177-178)

Figure Legends

Figure 1

(h) ChIP assay, using antibodies recognizing XBP-1 or ATF6 , for analysis of binding with the *Sdf211* promoter in the liver, in a 24-hour fasted state and a 3-hour refed state ($n = 3$). The relative expression levels were normalized by the upstream region (-3.5kb) and then by control IgG (rabbit or mouse IgG, respectively). (p. 41, ll. 922-924)

Figure 5

(d) ChIP assay, using antibodies recognizing XBP-1 or ATF6, for analysis of binding of transcription factors with the *Sdf211* promoter in the liver, in a 1-hour refed state after 24 hours of fasting ($n = 3$). The relative expression levels were normalized by the upstream region (-3.5kb) and then by control IgG (rabbit or mouse IgG, respectively). (p.44, ll. 993-997)

Supplementary Figure Legends

Supplementary Figure 3

(h) ChIP assay for analysis of binding with the *Hspa5* promoter, in a 24-hour fasted state and a 3-hour refed state ($n = 3$). The relative expression levels were normalized by the upstream region (-4.5kb) and then by control IgG. (p.2)

Supplementary Figure 7

(c) ChIP assay for analysis of binding with the *Hspa5* promoter, in a 1-hour refed state after 24 hours of fasting ($n = 3$). The relative expression levels were normalized by the upstream region (-4.5kb) and then by control IgG. (p.4)

Figure labeling

Figure 1h, 5d

Sdf211 ROI / -3.5kb -> *Sdf211* ROI

Supplementary Figure 3h, 5d

Hspa5 ERSE / -4.5kb -> *Hspa5* ERSE

> Additionally, the labeling and legends in the figures were poorly composed of.

We modified labeling in figures to show results of ChIP assays, as stated above (Figure 1h, 5d and Supplementary Figure 3h, 5d).

Figure 1 Only for the Reviewer

ChIP assay samples in the fasted state and the refed state of the liver in wild type mice were subjected to PCR, followed by agarose gel electrophoresis of, (a) plateau-phase products, and (b) log-phase products, respectively. The expected product size: 119bp for ROI of the *Sdf2l1* promoter, 239bp for ERSE of the *Hspa5* promoter.

REVIEWERS' COMMENTS:

Reviewer #1 (Remarks to the Author):

Comments addressed.